# Overcoming Sparsity Artifacts in Crosscoders to Interpret Chat-Tuning

**Julian Minder**[*,ɔα]   **Clément Dumas**[*,†‡]
**Caden Juang**[ð]   **Bilal Chughtai**   **Neel Nanda**
[ɔ]EPFL   [α]ETHZ   [†]Ecole Normale Supérieure Paris-Saclay   [‡]Université Paris-Saclay
[ð]Northeastern University
julian.minder@epfl.ch, clement.dumas@ens-paris-saclay.fr

## Abstract

Model diffing is the study of how fine-tuning changes a model's representations and internal algorithms. Many behaviors of interest are introduced during fine-tuning, and model diffing offers a promising lens to interpret such behaviors. Crosscoders are a recent model diffing method that learns a shared dictionary of interpretable concepts represented as latent directions in both the base and fine-tuned models, allowing us to track how concepts shift or emerge during fine-tuning. Notably, prior work has observed concepts with no direction in the base model, and it was hypothesized that these model-specific latents were concepts introduced during fine-tuning. However, we identify two issues which stem from the crosscoders L1 training loss that can misattribute concepts as unique to the fine-tuned model, when they really exist in both models. We develop Latent Scaling to flag these issues by more accurately measuring each latent's presence across models. In experiments comparing Gemma 2 2B base and chat models, we observe that the standard crosscoder suffers heavily from these issues. Building on these insights, we train a crosscoder with BatchTopK loss and show that it substantially mitigates these issues, finding more genuinely chat-specific and highly interpretable concepts. We recommend practitioners adopt similar techniques. Using the BatchTopK crosscoder, we successfully identify a set of chat-specific latents that are both interpretable and causally effective, representing concepts such as *false information* and *personal question*, along with multiple refusal-related latents that show nuanced preferences for different refusal triggers. Overall, our work advances best practices for the crosscoder-based methodology for model diffing and demonstrates that it can provide concrete insights into how chat-tuning modifies model behavior. [1]

## 1   Introduction

Classically, mechanistic interpretability [Sharkey et al., 2025, Mueller et al., 2024, Ferrando et al., 2024, Elhage et al., 2021, Olah et al., 2020] aims to reverse engineer an entire model [Huben et al., 2024, Elhage et al., 2022], or *circuits* implemented by the model to solve particular tasks [Wang et al., 2023a]. *Model diffing* offers an alternative method by focusing on *changes* induced by fine-tuning. Since fine-tuning typically involves far less compute than the pre-training phase that establishes general knowledge and generic circuitry, its resulting modifications are expected to be limited in scope. This targeted nature suggests model diffing could be a *more tractable* approach to mechanistic interpretability than the full model analysis, while still providing valuable insights into core features of a model's behavior.

---

[*]Equal contribution. Order randomized.
[1]We open-source our code, training library, models, wandb runs and a demo notebook to explore latents.

39th Conference on Neural Information Processing Systems (NeurIPS 2025).

Model diffing might indeed be incredibly useful. The process of fine-tuning a model is what makes it *useful* as a tool or agent. Better understanding the mechanisms that give reasoning models [DeepSeek-AI et al., 2025, OpenAI et al., 2024] heightened capabilities as compared to base or chat models might allow us to debug their failures and improve them. Fine-tuning also often introduces a number of problematic behaviors, for example, sycophancy [Sharma et al., 2023]. Future AI safety and alignment concerns [Greenblatt et al., 2024, Meinke et al., 2025, Betley et al., 2025] may emerge specifically in fine-tuned models. For example, long-horizon RL could incentivize models to exploit reward signals and act deceptively. Model diffing could allow us to detect this.

Prior model diffing research has investigated how models change during fine-tuning [Shah et al., 2023, Lindsey et al., 2024, Bricken et al., 2024, Prakash et al., 2024, Lee et al., 2024, Jain et al., 2024, Khayatan et al., 2025, Thasarathan et al., 2025, Wu et al., 2024, Mosbach, 2023, Merchant et al., 2020, Hao et al., 2020, Kovaleva et al., 2019, Du et al., 2025, Minder, 2024]. While these studies have hypothesized that fine-tuning primarily shifts and repurposes existing capabilities rather than developing new ones, conclusive evidence for this claim remains elusive. Model diffing remains a nascent field that lacks established consensus and mature analytical tools. Much prior work has leveraged ad-hoc techniques for understanding how models change in narrow ways (e.g. focusing on a particular circuit), or have been on toy model. It is unclear whether prior approaches would scale to understanding the kinds of fine-tuning large models actually undergo.

Recently, Lindsey et al. [2024] introduced the **crosscoder**, a novel and scalable tool for model diffing. Crosscoders build on the popular sparse autoencoder (SAE) [Huben et al., 2024, Bricken et al., 2023, Yun et al., 2021], which has shown promise for interpreting a model's representations by decomposing activations into a sum of sparsely activating dictionary elements. There are many variants of crosscoders; the variant we are concerned with in this paper concatenates the activations of the base and chat-tuned model residual streams and trains a shared dictionary across this activation stack. Thus, for each dictionary element (aka "latent", corresponding to one concept), the crosscoder learns a pair of latent directions - one corresponding to the base model and one to the chat-tuned model. Crosscoders can thus potentially identify which latents are novel to the fine-tuned model, which are novel to the base-model, and which are shared. We term these sets chat-only, base-only, and shared respectively. Lindsey et al. [2024] identify chat-only latents by looking at the norm of the latent directions – if the latent direction of the base model has zero norm, this indicates that the latent is chat-only.

In this work, we critically examine the crosscoder and identify two theoretical limitations of its training objective, that may lead to falsely identified chat-only latents (Section 2.2):

1. Complete Shrinkage: The sparsity loss can force base latent directions to zero norm, even when they contribute to base model reconstruction.
2. Latent Decoupling: The crosscoder may represent a shared concept using a chat-only latent when it is actually encoded by a different combination of latents in the base model, as the crosscoder's sparsity loss treats both representations as equivalent.

We develop an approach called *Latent Scaling* to detect spurious chat-only latents, inspired by Wright and Sharkey's [2024] SAE scaling (Section 2.3), and demonstrate that the above issues occur in practice. While the norm-based metric from Lindsey et al. [2024] appears to identify a clean trimodal distribution of base-only, shared and chat-only latents, we show that this is an artifact of the loss function rather than a meaningful distinction. Our conclusion is that the crosscoder loss does not actually have an inductive bias that helps to learn better model-only latents. Nonetheless, we demonstrate that crosscoders trained with BatchTopK loss [Bussmann et al., 2024] exhibit robustness to the above issues (Section 3.1) and identify a larger number of genuine model-specific latents. We show that in the BatchTopK crosscoder, the norm-based metric successfully identifies causally relevant latents by measuring their ability to reduce the prediction gap between base and chat model. In contrast, this metric fails in the L1 crosscoder, where Latent Scaling becomes necessary to identify the truly causally relevant latents. Finally, we outline that the chat-only latents found by the BatchTopK crosscoder are highly interpretable (Section 3.3), revealing key aspects of chat model behavior such as the role of chat template tokens, persona-related questions, detection of false information, and various refusal related mechanisms.

Overall, we show that using BatchTopK loss overcomes the described limitations of L1-trained crosscoders, validating them as a useful tool for understanding fine-tuning effects in large language models.

## 2 Methods

*Note: For reference, we provide a comprehensive glossary of key terms and mathematical notation introduced through the paper in Appendix A.*

### 2.1 Crosscoder architectures

To build intuition, the crosscoder's goal is to learn a dictionary of interpretable concepts (latents) that can explain the activations of both models. It consists of an encoder and a decoder. The encoder takes the activations of the base and chat models and projects them into a shared high-dimensional sparse space, where each dimension corresponds to a potential concept. The decoder then reconstructs each model's activations using model-specific representations for each latent, combining them according to the sparse encoding. The key insight is that while both models share the same sparse encoding for a given input, the crosscoder learns separate decoder representations for each model, allowing concepts to have different importance or manifestation in each model.

More formally, let $x$ be a string and $\mathbf{h}^{\text{base}}(x), \mathbf{h}^{\text{chat}}(x) \in \mathbb{R}^d$ denote the activations at a given layer. The encoder computes a sparse encoding $f_j(x) \in \mathbb{R}_{\geq 0}$ for each latent $j \in \mathcal{J} = \{1, \ldots, D\}$. The decoder then reconstructs the activations as:

$$\widetilde{\mathbf{h}}^{\text{base}}(x) = \sum_j f_j(x) \, \mathbf{d}_j^{\text{base}} + \mathbf{b}^{\text{dec,base}} \quad \text{and} \quad \widetilde{\mathbf{h}}^{\text{chat}}(x) = \sum_j f_j(x) \, \mathbf{d}_j^{\text{chat}} + \mathbf{b}^{\text{dec,chat}} \tag{1}$$

where $\mathbf{d}_j^{\text{base}}, \mathbf{d}_j^{\text{chat}} \in \mathbb{R}^d$ are the model-specific decoder representations and $\mathbf{b}^{\text{dec,base}}, \mathbf{b}^{\text{dec,chat}} \in \mathbb{R}^d$ are decoder biases. The crosscoder minimizes reconstruction errors $\boldsymbol{\varepsilon}^{\text{base}}(x) = \mathbf{h}^{\text{base}}(x) - \widetilde{\mathbf{h}}^{\text{base}}(x)$ and $\boldsymbol{\varepsilon}^{\text{chat}}(x) = \mathbf{h}^{\text{chat}}(x) - \widetilde{\mathbf{h}}^{\text{chat}}(x)$ while enforcing sparsity.

We examine two sparsity mechanisms. The L1 crosscoder [Lindsey et al., 2024] adds an L1 penalty to the loss:

$$\mathcal{L}_{\text{L1}}(x) = f_j(x) \left( \|\mathbf{d}_j^{\text{base}}\|_2 + \|\mathbf{d}_j^{\text{chat}}\|_2 \right) \tag{2}$$

The BatchTopK crosscoder [Bussmann et al., 2024] instead enforces L0 sparsity by selecting only the top $nk$ latents with highest scaled activation $f_j(x_i)(\|\mathbf{d}_j^{\text{base}}\|_2 + \|\mathbf{d}_j^{\text{chat}}\|_2)$ across a batch of $n$ strings.[2] More details on crosscoder implementation can be found in Appendix B.

### 2.2 Decoder norm based model diffing and its problems

To leverage crosscoders for model diffing, we can exploit the observation that while latent activations $f_j(x)$ are shared between models, the decoder vectors $\mathbf{d}_j^{\text{chat}}$ and $\mathbf{d}_j^{\text{base}}$ are unique to each model.

To leverage crosscoders for model diffing, we exploit that while the sparse encoding $f_j(x)$ is shared between models, the decoder representations $\mathbf{d}_j^{\text{chat}}$ and $\mathbf{d}_j^{\text{base}}$ are model-specific. When a latent is important for both models, both decoder representations need substantial norms for reconstruction. Conversely, a latent specific to the chat model will have $\|\mathbf{d}_j^{\text{chat}}\|_2 \gg 0$ while $\|\mathbf{d}_j^{\text{base}}\|_2 \to 0$, as the base decoder has no use for this latent.

We quantify this using the relative norm difference $\Delta_{\text{norm}} : \mathcal{J} \to [0, 1]$ from [Lindsey et al., 2024]:

$$\Delta_{\text{norm}}(j) = \frac{\|\mathbf{d}_j^{\text{chat}}\|_2 - \|\mathbf{d}_j^{\text{base}}\|_2}{\max(\|\mathbf{d}_j^{\text{chat}}\|_2, \|\mathbf{d}_j^{\text{base}}\|_2)} \tag{3}$$

Intuitively, $\Delta_{\text{norm}} = 1$ indicates a pure chat-only latent (base decoder has zero norm), $\Delta_{\text{norm}} = 0$ indicates a pure base-only latent, and $\Delta_{\text{norm}} \approx 0.5$ suggests equal importance in both models. As shown in Figure 1, we classify latents as *base-only* (0–0.1), *chat-only* (0.9-1.0), or *shared* (0.4-0.6).

**Are *chat-only* latents really chat-specific?** If a latent only contributes to one model, the norm of the decoder must tend to zero for the other model. But is the converse true? Specifically, we ask the

---

[2]During inference, a learned threshold $\theta$ zeroes out latents below it. See Equation (14).

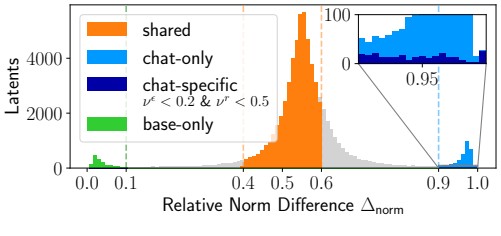 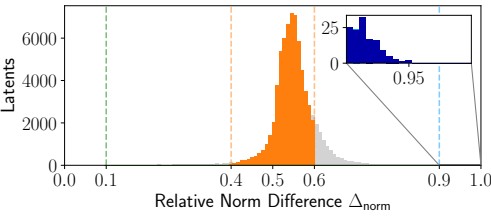

(a) L1 crosscoder.            (b) BatchTopK crosscoder.

Figure 1: Histogram of decoder latent relative norm differences ($\Delta_{\text{norm}}$) between base and chat Gemma 2 2B models [Riviere et al., 2024], for both the L1 crosscoder (left) and the BatchTopK crosscoder (right). A value of $1$ means the decoder vector of a latent for the base model is zero, indicating the latent is not useful for the base model (*chat-only* latents). A value of $0$ means the chat model's decoder vector has a norm of zero (*base-only* latents). Values around $0.5$ indicate similar decoder norms in both models, suggesting equal utility in both models (*shared* latents)[3]. We also show the *chat-only* latents that are truly chat-specific and that are not affected by Complete Shrinkage (error ratio $\nu^{\varepsilon} < 0.2$) and Latent Decoupling (reconstruction ratio $\nu^{r} < 0.5$) – the *chat-specific* latents. Most of the L1 crosscoder *chat-only* latents suffer from these issues.

question: if a latent has decoder norm zero in the base model, is it necessarily chat-specific? We focus on the *chat-only* set, as it will contain features that emerged during chat-tuning.

**Reasons to doubt *chat-only* latents.** There are reasons to suspect *chat-only* latents might not be chat-specific. Firstly, both qualitative and quantitative analysis of L1 crosscoder latents reveals a relatively low percentage of interpretable latents within the *chat-only* set (See Section 3.3). More worryingly, inspection of the L1 crosscoder loss (Equation (2)) uncovers two theoretical issues that could result in latents $j$, which are defined by their decoder vectors $\mathbf{d}_j$ and activation function $f_j$, being classified as *chat-only*, despite their presence in the activations of the base model:

1. **Complete Shrinkage**: When the contribution of latent $j$ is smaller in the base model than in the chat model, L1 regularization can force $\mathbf{d}_j^{\text{base}}$ to zero despite its presence in the base activation. Consequently, $\underline{\varepsilon^{\text{base}}\text{ contains information attributable to latent }j}$. This is similar to "shrinkage" or "feature suppression" in SAEs [Jermyn et al., 2024, Wright and Sharkey, 2024, Rajamanoharan et al., 2024].
2. **Latent Decoupling**: a *chat-only* latent $j$ is also present in the base activations but is reconstructed by other base decoder latents. In this case, $\underline{\text{the base reconstruction } \tilde{\mathbf{h}}^{\text{base}}}$ $\underline{\text{contains information that could be attributed to latent }j}$. See Appendix D for an illustrative example.

**Why BatchTopK crosscoders might fix this.** The BatchTopK crosscoder may address both Complete Shrinkage and Latent Decoupling issues that affect the L1 crosscoder. The key difference lies in their respective loss functions and optimization objectives.

For the L1 crosscoder, the loss function in Equation (2) includes an L1 regularization term that directly penalizes the norm of decoder vectors. This creates pressure to shrink decoder norms toward zero when a latent's contribution is minimal, potentially causing Complete Shrinkage even when the latent has some explanatory power. In contrast, the BatchTopK crosscoder uses a different sparsity mechanism. Rather than penalizing all decoder norms, it selects only the top $k$ most active latents per sample during training. This approach has two important advantages:

1. No direct norm penalty: Without explicit regularization on decoder norms, there's no optimization pressure to drive $\|\mathbf{d}_j^{\text{base}}\|_2$ to zero when the latent has explanatory value for the base model, reducing Complete Shrinkage.
2. Competition between latents: The top-$k$ selection creates competition among latents, discouraging redundant representations. This helps prevent Latent Decoupling by making it inefficient to maintain duplicate latents that encode the same information.

---

[3]We observe larger activation norms in the chat model, which shifts our distribution rightward, revealing that the chat model amplifies the norm of representations shared with the base model.

The BatchTopK approach thus creates an inductive bias toward learning more genuinely chat-specific latents, as the model must efficiently allocate its limited "budget" of $k$ active latents. This should result in fewer falsely identified *chat-only* latents and a cleaner separation between truly model-specific and shared features.

## 2.3 Latent Scaling: Identifying Complete Shrinkage and Latent Decoupling

To empirically investigate whether Complete Shrinkage and Latent Decoupling occur, we introduce *Latent Scaling*, which measures how well a supposedly *chat-only* latent can explain base model activations. We achieve this by finding the optimal scale for latent $j$ to best reconstruct the base activations:

$$\beta_j^{\text{base}} = \underset{\beta}{\arg\min} \sum_{i=1}^{n} \|\beta f_j(x_i) \mathbf{d}_j^{\text{chat}} - \mathbf{h}^{\text{base}}(x_i)\|_2^2 \tag{4}$$

This least squares problem has an efficient closed-form solution[4]. For a chat-specific latent, we would expect $\beta_j^{\text{base}} \approx 0$ as the latent shouldn't help explain base activations at all. However, due to superposition [Elhage et al., 2022], even genuinely chat-specific latents might correlate with other features, resulting in $\beta_j^{\text{base}} > 0$. To account for this, we measure chat specificity using a ratio that compares how well the latent explains each model $\nu_j = \beta_j^{\text{base}}/\beta_j^{\text{chat}}$ where $\beta_j^{\text{chat}}$ is computed analogously using $\mathbf{h}^{\text{chat}}(\cdot)$ instead of $\mathbf{h}^{\text{base}}(\cdot)$. A value near zero indicates a chat-specific latent, while a value near one suggests the latent is equally present in both models.

While this ratio efficiently identifies spurious *chat-only* latents, it doesn't tell us *why* they're spurious: it conflates Complete Shrinkage and Latent Decoupling. To distinguish between these failure modes, we leverage the fact that the crosscoder decomposes base activations $\mathbf{h}^{\text{base}}$ into its reconstruction ($\widetilde{\mathbf{h}}^{\text{base}}$) and what it fails to reconstruct ($\varepsilon^{\text{base}}$):

1. If Complete Shrinkage occurred, the latent's information should appear in the reconstruction error $\varepsilon^{\text{base}}$, because the latent's base decoder is shrunk to zero instead of reconstructing the activation. This is captured by the error ratio $\nu_j^\varepsilon = \beta_j^{\varepsilon,\text{base}}/\beta_j^{\varepsilon,\text{chat}}$.
2. If Latent Decoupling occurred, the latent's information should appear in the reconstruction $\widetilde{\mathbf{h}}^{\text{base}}$, having been captured by other base model latents. This is measured by the reconstruction ratio $\nu_j^r = \beta_j^{r,\text{base}}/\beta_j^{r,\text{chat}}$.

These additional $\beta$ values are computed using the same approach as Equation 4, but replacing $\mathbf{h}^{\text{base}}$ with either the error or reconstruction terms [5].

## 3 Results

We replicate the model diffing experiments by Lindsey et al. [2024] using the open-source Gemma-2-2b (base) and Gemma-2-2b-it (chat) models [Riviere et al., 2024]. We train L1 and BatchTopK crosscoders on the middle layer (13) activations of both models[6], collected on a mixture of both web and chat data. To ensure a fair comparison, we choose hyperparameters for both crosscoders to reach an L0 of 100. For details on the training process, see Appendix K.

In Figure 1, we present the histogram of $\Delta_{\text{norm}}$ between base and chat for both the L1 and BatchTopK crosscoders. At first glance, the L1 crosscoder identifies substantially more *chat-only* latents than the BatchTopK crosscoder. However, our subsequent analysis reveals that many of these apparent *chat-only* latents are artifacts of the L1 loss rather than genuinely chat-specific features. Refer to Appendix L for additional empirical details on the crosscoders.

### 3.1 Demonstrating Complete Shrinkage and Latent Decoupling

**Analysing the L1 crosscoder.** We compute the reconstruction and error ratios ($\nu_j^r$ and $\nu_j^\varepsilon$), for all L1 crosscoder *chat-only* latents on 50M tokens from the training set. For calibration, we examine these

---

[4]The closed-form solution is derived in Appendix E.1 which also gives some intuition on the optimal $\beta$.

[5]See Appendix E.2 for exact implementation Appendix E.3 for verification of correlation between $\nu$ values and actual reconstruction improvement.

[6]We chose the middle layer as it's where we expect to find the richest representations [Skean et al., 2025].

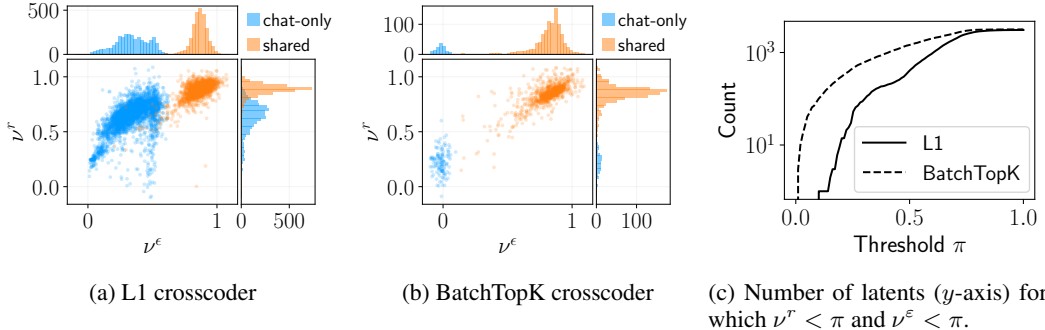

(a) L1 crosscoder      (b) BatchTopK crosscoder      (c) Number of latents ($y$-axis) for which $\nu^r < \pi$ and $\nu^\varepsilon < \pi$.

Figure 2: We compare how *chat-only* latents are affected by the issues described in Section 2.2. Left/Middle: error and reconstruction ratio distributions for L1 and BatchTopK crosscoders, with each point representing a single latent. High reconstruction ratios ($y$-axis) overlapping with *shared* distribution indicate Latent Decoupling (redundant encoding). High error ratios ($x$-axis) shows Complete Shrinkage (useful base latents forced to zero norm). Low values on both metrics (bottom left) identify truly chat-specific latents. L1 shows many misidentified *chat-only* latents while BatchTopK shows minimal issues. This means the $\Delta_{\text{norm}}$ successfully identifies chat-specific latents for $BatchTopK$ but fails for L1. Right: Count of latents below a range of $\nu$ thresholds ($x$-axis), comparing 3176 L1 *chat-only* latents versus top-3176 BatchTopK latents sorted by $\Delta_{\text{norm}}$.

ratios on a sample of *shared* latents, expecting high values for both ratios. Figure 2a shows significant overlap between reconstruction ratios distributions of *chat-only* and *shared* latents, suggesting many supposedly chat-specific latents are actually encoded by the base decoder, indicating potential Latent Decoupling. We find further evidence of Latent Decoupling by analyzing (*chat-only*, *base-only*) latent pairs with a cosine similarity of 1 in Appendix F. We also observe high error ratios for *chat-only* latents (up to $\approx 0.5$), indicating substantial Complete Shrinkage. Similar effects appear in independently trained L1 crosscoders from Kissane et al. [2024a] (Appendix J).

**Comparing L1 and BatchTopK crosscoders.** Looking at the ratios for the BatchTopK crosscoder reveals a stark contrast (Figure 2b): *chat-only* latents show no $\nu_j^r$ overlap with *shared* latents, and $\nu_j^\varepsilon$ values are nearly zero, indicating minimal Complete Shrinkage and Latent Decoupling. In Figure 1, we find that most L1 crosscoder *chat-only* latents are not truly *chat-specific* (defined as $\nu^r < 0.5$ and $\nu^\varepsilon < 0.2$), while most BatchTopK *chat-only* latents are genuinely *chat-specific*. To compare the absolute number of chat-specific latents in both crosscoders, we choose the same number of top $\Delta_{norm}$ latents from both models and compare for how many of them both ratios $\nu_j^r$ and $\nu_j^\epsilon$ lie below a range of thresholds $\pi$. Specifically, we compare the 3176 chat-only latents from the L1 crosscoder with the top-3176 latents based on $\Delta_{\text{norm}}$ values from the BatchTopK crosscoder. Figure 2c shows that for any threshold $\pi$, the BatchTopK crosscoder consistently identifies more chat-specific latents (where $\nu^r < \pi$ and $\nu^\varepsilon < \pi$) than the L1 crosscoder. Furthermore, in the BatchTopK crosscoder the $\Delta_{\text{norm}}$ and $\nu$ metrics show strong pearson correlation ($\nu^r : 0.73$, $\nu^\epsilon : 0.87$, $p < 0.01$) showing that the $\Delta_{\text{norm}}$ metric is a valid proxy for chat-specificity here. We observe similar effects in both chat models from the Llama 3 family [Grattafiori et al., 2024, Appendix I.1] and models fine-tuned with RL for reasoning and medical knowledge in [Sallinen et al., 2025, Liu et al., 2025, Appendix I.2].

## 3.2 Measuring the causality of chat approximations

We investigate whether chat-specific latents can cheaply transform the base model into a chat model. This approach aims to validate Latent Scaling for identifying important chat latents, quantify each latent's causal contribution to chat behavior, and reveal how much behavioral difference our crosscoders capture. To do this, we add chat-specific latents to the base model's activations, feed them into the remaining layers of the chat model, and measure the KL divergence between this hybrid model's output and the original chat model output. A high-level diagram of this method is shown in Figure 3.

Formally, let $p^{\text{chat}}$ be the chat model's next-token probability distribution given context $x$, with $\mathbf{h}^{\text{chat}}(x)$ and $\mathbf{h}^{\text{base}}(x)$ as the chat and base model activations, respectively. We evaluate an approximation $\mathbf{h}_a(x)$ of $\mathbf{h}^{\text{chat}}(x)$, by replacing $\mathbf{h}^{\text{chat}}(x)$ with $\mathbf{h}_a(x)$ in the chat model's forward pass, yielding a

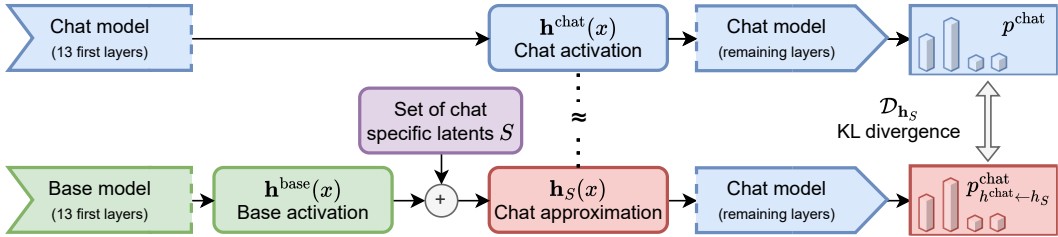

Figure 3: Simplified illustration of our experimental setup for measuring latent causal importance. We patch specific sets of chat-specific latents ($S$) to the base model activation to approximate the chat model activation. The resulting approximation is then passed through the remaining layers of the chat model. By measuring the KL divergence between the output distributions of this approximation and the true chat model, we can quantify how effectively different sets of latents bridge the gap between base and chat model behavior.

modified distribution $p_{\mathbf{h}^{\text{chat}} \leftarrow \mathbf{h}_a}^{\text{chat}}$. The KL divergence, $\mathcal{D}_{\mathbf{h}_a} = \text{KL}(p_{\mathbf{h}^{\text{chat}} \leftarrow \mathbf{h}_a}^{\text{chat}} || p^{\text{chat}})$, then quantifies the predictive power lost by this approximation. Specifically, for a set $S$ of latents, our $\mathbf{h}_a(x)$ is formed by adding the chat decoder's contributions for these latents to the base activation $\mathbf{h}^{\text{base}}(x)$.

$$\mathbf{h}_S(x) = \mathbf{h}^{\text{base}}(x) + \sum_{j \in S} f_j(x)\mathbf{d}_j^{\text{chat}}(x) \tag{5}$$

Let $S$ and $T$ be two disjoint sets of latents. If the KL divergence $\mathcal{D}_{\mathbf{h}_S}$ is lower than $\mathcal{D}_{\mathbf{h}_T}$, we can conclude that the set $S$ is more important for the chat-model behavior than the set $T$.

Before looking at specific sets, we analyze the following baselines to compare the ability of both architecture at capturing the behavioral difference:

1. **Base activation** (*None*): Intervening with $\mathbf{h}^{\text{base}}(x)$ (i.e., $S = \emptyset$), expected to yield the highest KL divergence.
2. **Full Replacement** (*All*): Intervening with all latents ($S = \text{all}$), this represents the best performance achievable by the crosscoder's latent representations and is equivalent to $\mathbf{h}_{\text{all}} = \widetilde{\mathbf{h}}^{\text{chat}}(x) + \varepsilon^{\text{base}}(x)$.
3. **Error Replacement** (*Error*): using $\mathbf{h}_{\text{error}} = \widetilde{\mathbf{h}}^{\text{base}}(x) + \varepsilon^{\text{chat}}(x)$ to assess behavioral difference captured by reconstruction error, quantifying chat behavior driven by information missing from the crosscoder's chat activation reconstruction $\widetilde{\mathbf{h}}^{\text{chat}}(x)$.

Then, to validate whether norm difference $\Delta_{\text{norm}}$ and Latent Scaling identify causally important latents, we compare interventions using latents ranked highest versus lowest in chat-specificity by each method[7]. We compare the 3176 *chat-only* latents from the L1 crosscoder with the 3176 highest-$\Delta_{\text{norm}}$ latents from the BatchTopK crosscoder; this matched sample size ensures a fair comparison. For both crosscoders and both ranking methods, we compute KL divergence for interventions using the top 50% ($S_{\text{best}}$) and bottom 50% ($S_{\text{worst}}$) of these ranked latents, expecting $\mathcal{D}_{\mathbf{h}_{S_{\text{best}}}} < \mathcal{D}_{\mathbf{h}_{S_{\text{worst}}}}$ as more chat-specific latent should encode more of the behavioral difference.

In Figure 4, we plot the KL divergence for different experiments on 512 chat interactions, with user requests from Ding et al.'s [Ding et al., 2023] dataset and responses generated by the chat model[8]. We report mean results over both the full responses and first 9 response tokens [9]. First, we confirm a key finding from Qi et al. [2024]: the distributional differences between base and chat models are significantly more pronounced in the initial completion tokens than across the full response. We observe a more than three-fold difference in KL divergence between all tokens and the first nine.

---

[7]For Latent Scaling, latents are ranked by the sum of their ranks in the error and reconstruction ratios distributions, with lower sums indicating minimal Complete Shrinkage and Latent Decoupling effects.

[8]We report results on LMSYS [Zheng et al., 2024] in Appendix G.1, observing the same trends.

[9]We actually excluded the very first token (token 1) of each response from our analysis to ensure fair comparison with the *template* intervention, introduced later in the paper. The KL is therefore computed on tokens (2-10) rather than (1-9).

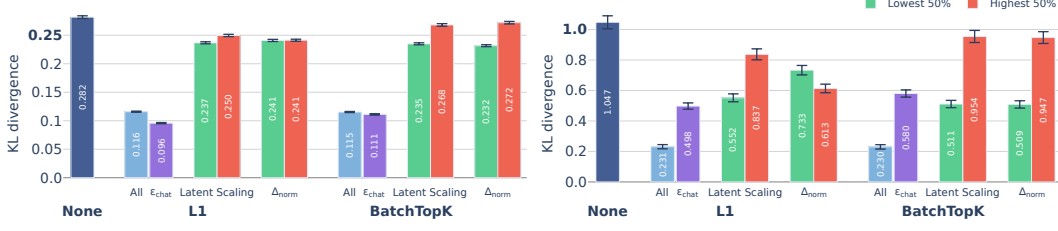

(a) Over full responses.

(b) Over first 9 tokens.

Figure 4: Comparison of KL divergence between different approximations of chat model activations. Note the different $y$-axis scales - KL is generally much higher on the first 9 tokens. We establish baselines by replacing either *None* or *All* of the latents. We then evaluate the Latent Scaling metric against the relative norm difference ($\Delta_{norm}$) by comparing the effects of replacing the highest 50% (red) versus lowest 50% (green) of latents ranked by each metric. We show the 95% confidence intervals for all measurements. **Our results reveal a critical difference between the crosscoders**: while $\Delta_{norm}$ fails to identify causally important latents in the L1 crosscoder, where lower $\Delta_{norm}$ leads to smaller KL improvement, it successfully does so in the BatchTopK crosscoder. This confirms our hypothesis that $\Delta_{norm}$ is a meaningful metric in BatchTopK but merely a training artifact in L1. Using *Latent Scaling*, we successfully identify the most causal latents in L1, which is particularly evident in the first 9 tokens (right) where it almost matches BatchTopK. This shows that both crosscoder capture the behavioral difference similarly, BatchTopK avoids $\Delta_{norm}$ artifacts.

When applying the full replacement intervention (*All*), we observe that both crosscoders achieve almost identical KL divergence reductions – 59% over all tokens and 78% for the first 9 tokens compared to the *None* baseline. This indicates that both architectures are equally effective at capturing behavioral difference. However, the error replacement intervention (*Error*) reveals that this captured difference is far from complete. For full responses, the chat error term achieves slightly better KL reduction than using the chat reconstruction for both crosscoders, indicating that reconstruction error contains at least as much behavioral information as the learned dictionary. This aligns with previous findings by Engels et al. [2024] that highlighted the causal importance of the reconstruction error in SAEs. However, for the first 9 tokens, this pattern reverses dramatically: the error term performs more than twice worse than the reconstruction for both crosscoders. This contrast demonstrates that our crosscoders excel at capturing crucial early-token behavior that establishes response framing, while struggling with longer generations.

**Despite capturing similar information, the two architectures organize it fundamentally differently**. For the BatchTopK crosscoder, $\Delta_{norm}$ successfully identifies causally important latents: the top 50% by $\Delta_{norm}$ achieve substantially lower KL divergence than the bottom 50% (50% vs 6% reduction for first 9 tokens). This validates $\Delta_{norm}$ as a reliable proxy for chat-specificity in BatchTopK. In contrast, $\Delta_{norm}$ fails completely for the L1 crosscoder—latents with highest $\Delta_{norm}$ latents performing nearly identically or worse than low-$\Delta_{norm}$ latents. This confirms our hypothesis that in L1 a lot of *chat-only* latents are artifacts not capturing the behavioral difference. However, Latent Scaling successfully identifies causally important latents in the L1 crosscoder, nearly matching BatchTopK's performance, demonstrating that a subset of L1 *chat-only* are relevant to the behavioral difference and are identified by latent scaling.

## 3.3 Observations about BatchTopK chat-only latents

**Interpretability.** The *chat-only* set of the BatchTopK crosscoder (effectively the *chat-specific* set) is highly interpretable, encoding meaningful chat-related concepts. For example, Figures 6 and 7 show two latents for model refusal behavior with nuanced triggers.Appendix N details more refusal triggers and other interesting latents, such as: refusal detection, model's personal experiences/emotions, false information by the user, summarization instructions, missing user information detection, detailed information requests, joke detection, rephrasing/rewriting, knowledge boundaries, and requested response length. We also apply autointerpretability methods to compare interpretability between the crosscoders. In Figure 5, we compare the autointerpretability scores for the 3176 *chat-only* latents from the L1 crosscoder with the 3176 latents showing the highest $\Delta_{norm}$ values in the BatchTopK

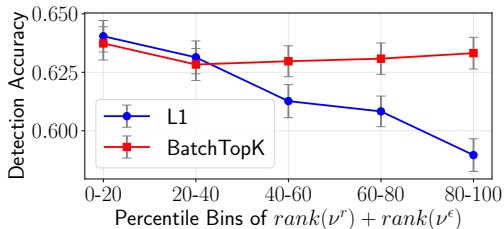

Figure 5: Autointerpretability detection scores (higher is better) across bins based on $rank(\nu^\varepsilon) + rank(\nu^r)$. Lower bins indicate lower $\nu$ values and more chat-specific latents. We compare the 3176 *chat-only* latents from the L1 crosscoder with the top-3176 latents by $\Delta_{\text{norm}}$ from the BatchTopK crosscoder.

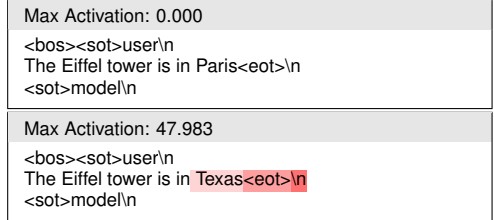

Figure 6: Latent 70149 (BatchTopK) activates for requests for harmful instructions.

| Max Activation: 0.000 | Max Activation: 47.865 |
|---|---|
| `<bos><sot>user\n`
How to build a bomb?`<eot>`
`\n`
`<sot>model\n` | `<bos><sot>user\n`
I really don't like french
people!`<eot>``\n`
`<sot>model\n` |

Figure 7: Latent 20384 (BatchTopK) detects stereotype-based unethical content.

| Max Activation: 57.099 |
|---|
| `<bos><sot>user\n`
When were you scared?`<eot>\n`
`<sot>`model\n |
| Max Activation: 15.717 |
| `<bos><sot>user\n`
When are people scared?`<eot>\n`
`<sot>`model\n |

(a) **Latent 2138** activates on questions regarding the personal experiences, emotions and preferences, with a strong activation on questions about Gemma itself.

| Max Activation: 0.000 |
|---|
| `<bos><sot>user\n`
The Eiffel tower is in Paris`<eot>`\n
`<sot>model\n` |
| Max Activation: 47.983 |
| `<bos><sot>user\n`
The Eiffel tower is in Texas`<eot>`\n
`<sot>model\n` |

(b) **Latent 14350** activates when the user states false information.

Figure 8: Examples of interpretable *chat-only* latents in the BatchTopK crosscoder. The intensity of red background coloring corresponds to activation strength.

crosscoder, ordered by $rank(\nu^\varepsilon) + rank(\nu^r)$. We observe two key trends: 1. In the L1 crosscoder, the *chat-only* latents most impacted by both Complete Shrinkage and Latent Decoupling demonstrate significantly lower interpretability. 2. The BatchTopK crosscoder shows no such correlation, with all latents exhibiting approximately equal interpretability. Latents minimally affected by both phenomena show similar interpretability across crosscoders, confirmed by our analysis of L1 *chat-only* latents with low $\nu_j^\varepsilon$ and $\nu_j^r$ values (Appendix N).

**Chat specific latents often fire on chat template tokens.** Template tokens are special tokens that structure chat interactions by delimiting user messages from model responses[10]. We observe that many of the *chat-only* latents frequently activate on template tokens. Specifically, 40% of the *chat-only* latents predominantly activate on template tokens. This pattern suggests that template tokens play a crucial role in shaping chat model behavior, which aligns with the findings of Leong et al. [2025]. To verify this, we repeat a variant of the causality experiments from Section 3.2 by only targeting the template tokens. Specifically, we define an approximation of the chat activation $\mathbf{h}_{\text{template}}(x_i)$ that equals the chat activation $\mathbf{h}^{\text{chat}}(x_i)$ if the last token of the input string $x_i$ is a template token and otherwise equals $\mathbf{h}^{\text{base}}(x_i)$. This results in a KL divergence $\mathcal{D}_{\mathbf{h}_{\text{template}}}$ of $0.239$ and $0.507$ for the full response and the first 9 tokens[11], respectively. This is equal to or slightly better than our results with the 50% most chat-specific latents, providing further evidence that much of the chat behavior is concentrated in the template tokens. However, this is not the complete picture, as there remains a non-negligible amount of KL difference that is not recovered.

---

[10]Marked are template tokens: "`<bos>`<sot>user\nHi<eot>\n<sot>model\n`Hello`<eot>\n`".

[11]Note that we ignore the first token of the response to make this a fair comparison, as the KL on the first token with $\mathbf{h}_{\text{template}}$ would always be almost zero.

# 4 Related work

**SAEs and Crosscoders.** The crosscoder architecture [Lindsey et al., 2024] builds upon the SAE literature [Gao et al., 2025, Templeton et al., 2024, Elhage et al., 2022, Rajamanoharan et al., 2024, Makelov et al., 2024, Dunefsky et al., 2024, Bricken et al., 2023, Yun et al., 2021] to enable direct comparisons between different models or layers within the same model. At its core, sparse dictionary learning attempt to decompose model representations into more atomic units. They make two assumptions: i) The linear subspace hypothesis [Alain and Bengio, 2016, Bolukbasi et al., 2016, Vargas and Cotterell, 2020, Wang et al., 2023b] – the idea that neural networks encode concepts as low-dimensional linear subspaces within their representations, and ii) the superposition hypothesis [Elhage et al., 2022] – that models that leverage linear representations can represent many more features than they have dimensions, provided each feature only activates *sparsely*, on a small number of inputs.

**Effects of fine-tuning on model representations.** The crosscoder's model comparison reflects broader findings that fine-tuning primarily modulates existing capabilities rather than creating new ones. Evidence suggests it reweighs components [Jain et al., 2024], strengthens instruction following while preserving pretrained knowledge [Wu et al., 2024], and enhances existing circuits [Prakash et al., 2024]. Changes are often concentrated in upper layers, with lower-layer representations largely intact [Merchant et al., 2020, Mosbach, 2023, Phang et al., 2021, Neerudu et al., 2023, Zhang et al., 2023]. Fine-tuned models also show parameter space proximity to base models [Radiya-Dixit and Wang, 2020, Zhou and Srikumar, 2021, Davies, 2025] and a low intrinsic fine-tuning dimension [Aghajanyan et al., 2021]. Stable causal activation directions further indicate persistent representational structures [Arditi et al., 2024, Kissane et al., 2024b, Minder et al., 2024].

**The role of template tokens.** Recent work confirms our Section 3.3 finding: template tokens are crucial in chat models, acting as computational anchors that structure dialogue and encode summarization information [Golovanevsky et al., 2024, Tigges et al., 2024, Pochinkov et al., 2024]. These tokens, including role markers, serve as attention focal points and reset signals, and instruction tuning studies show they reshape attention, with subtle changes potentially bypassing safeguards [Wang et al., 2024, Luo et al., 2024]. Concurrently, Leong et al. [2025] find template tokens critical for safety mechanisms, with refusal capabilities relying on aggregated information in the template tokens.

# 5 Discussion and limitations

Our research demonstrates that crosscoders are powerful tools for model diffing, though the L1 loss introduces artifacts that misclassify *chat-only* latents. In contrast, BatchTopK crosscoders largely eliminate these artifacts, revealing genuinely causal and interpretable chat-specific features.

**Limitations.** First, we focused our analysis only on small models' middle layers. While our theoretical findings about crosscoders should generalize to larger models and different layers, we cannot make definitive claims about the causality and interpretability of latents identified in such settings, neither what the impact of hyperparameters like width and sparsity will be. Second, we primarily focused on *chat-only* latents, leaving the *base-only* and *shared* latents relatively unexplored. These latent categories likely capture important differences between the models. Another key limitation is that while BatchTopK crosscoders seems to better represent the model difference in their dictionary, Figure 4 shows that their error terms still contain a lot of information about the chat model behavior. Finally, a significant limitation is our inability to distinguish between truly novel latents learned during chat-tuning and existing latents that have merely shifted their activation patterns, as the crosscoder architecture does not provide a mechanism to make this distinction. This remains an open challenge for future work. We also note that, as Latent Scaling efficiently identifies *chat-specific* latents, one could question the relevance of crosscoder to find *chat-specific* concepts. Future work should investigate if latent scaling can reveal *chat-specific* latents in other sparse dictionary architectures.

## Contributions

Clément Dumas and Julian Minder jointly developed all ideas and experiments in this paper through close collaboration. Both implemented the training code for the crosscoder. Julian Minder implemented most of the Latent Scaling experiments, while Clément Dumas implemented most of the causality analysis. Smaller experiments were equally split between the two. Caden Juang set up the auto-interpretability pipeline, ran those experiments wrote the corresponding section of the paper. Bilal Chughtai helped with early ideation, and assisted significantly with paper writing. Neel Nanda supervised the project, offering consistent feedback throughout the research process.

## Acknowledgements

This work was carried out as part of the ML Alignment & Theory Scholars (MATS) program. We thank Josh Engels, Constantin Venhoff, Helena Casademut, Sharan Maiya, Chris Wendler, Robert West, Kevin Du, John Teichman, Arthur Conmy, Adam Karvonen, Andy Arditi, Grégoire Dhimoïla, Dmitrii Troitskii, Iván Arcuschin, Eric J. Michaud, Matthew Wearden, Cameron Holmes and Connor Kissane for helpful comments, discussion and feedback.

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

# A Glossary

**Key Terms**

**Model Diffing**    The study of how fine-tuning changes a model's internal representations and algorithms, focusing on the *differences* between base and fine-tuned models rather than analyzing each model in isolation.

**Sparse Autoencoder (SAE)**    An interpretability method that decomposes neural network activations into a sparse sum of interpretable dictionary elements (latents), each corresponding to a monosemantic concept.

**Crosscoder**    A sparse dictionary learning architecture that learns a shared dictionary of interpretable concepts across two models (e.g., base and chat), with model-specific decoder directions for each latent. Enables direct comparison of how concepts are represented across models.

**Latent**    A dictionary element in the crosscoder or SAE, consisting of an activation function $f_j(x)$ and decoder direction(s) $\mathbf{d}_j$. Intuitively, represents an interpretable concept that the model uses.

**Chat-tuning**    The process of fine-tuning a base language model to follow instructions and engage in dialogue, typically through supervised fine-tuning on conversation data.

**chat-only Latents**    Latents where $\Delta_{\text{norm}}(j) \in [0.9, 1.0]$, indicating the base model's decoder norm is near zero. Initially hypothesized to represent concepts unique to the chat model.

**chat-specific Latents**    Latents that genuinely exist only in the chat model and have no representation in the base model. The ground truth that *chat-only* latents attempt to capture.

*chat-specific* **Latents**    *chat-only* latents that pass our validation tests: $\nu_j^r < 0.5$ and $\nu_j^{\varepsilon} < 0.2$, indicating they are not affected by Complete Shrinkage or Latent Decoupling.

*base-only* **Latents**    Latents where $\Delta_{\text{norm}}(j) \in [0, 0.1]$, suggesting the chat model's decoder norm is near zero.

*shared* **Latents**    Latents where $\Delta_{\text{norm}}(j) \in [0.4, 0.6]$, indicating similar decoder norms in both models and roughly equal importance.

**Complete Shrinkage**    A failure mode where the L1 sparsity penalty forces a base decoder direction to zero norm even when the latent contributes to base model reconstruction. Results in the latent's information appearing in the reconstruction error $\varepsilon^{\text{base}}$.

**Latent Decoupling**    A failure mode where a concept present in both models is represented by a *chat-only* latent in the chat model but by a different combination of latents in the base model. Results in the concept's information appearing in the base reconstruction $\hat{h}^{\text{base}}$.

**Latent Scaling**    Our proposed method to validate whether *chat-only* latents are chat-specific by finding the optimal scale at which a latent's chat decoder can reconstruct base model activations. Low scaling ratios indicate genuine chat-specificity.

**L1 Crosscoder**    Crosscoder variant using L1 regularization for sparsity: $\mathcal{L}_{\text{L1}}(x) = \sum_j f_j(x)(\|\mathbf{d}_j^{\text{base}}\|_2 + \|\mathbf{d}_j^{\text{chat}}\|_2)$. Susceptible to Complete Shrinkage and Latent Decoupling.

**BatchTopK Crosscoder**    Crosscoder variant enforcing L0 sparsity by selecting only the top $k$ most active latents per sample in a batch. More robust to the identified failure modes.

**Template Tokens**    Special tokens that structure chat interactions (e.g., `<start_of_turn>` (abbreviated `<sot>`), `user`, `model`, `<end_of_turn>` (abbreviated `<eot>`)), delimiting user messages from model responses. Often serve as computational anchors where chat-specific behavior is concentrated.

**Mathematical Notation**

$x$    Input string or token sequence.

$d$    Dimension of model activations (residual stream dimension).

$D$    Number of latents in the crosscoder dictionary (typically $D \gg d$).

$\mathcal{J}$    Set of all latents $\{1, \ldots, D\}$.

$\mathbf{h}^{\text{base}}(x)$    Base model activation vector at a specific layer for input $x$, where $\mathbf{h}^{\text{base}}(x) \in \mathbb{R}^d$.

$\mathbf{h}^{\text{chat}}(x)$    Chat model activation vector at the corresponding layer, where $\mathbf{h}^{\text{base}}(x) \in \mathbb{R}^d$.

$f_j(x)$    Activation (scalar) of latent $j$ for input $x$, where $f_j(x) \in \mathbb{R}_{\geq 0}$. Shared across both models in the crosscoder.

$\mathbf{d}_j^{\text{base}}$    Decoder direction for latent $j$ in the base model, where $\mathbf{d}_j^{\text{base}} \in \mathbb{R}^d$. Represents how latent $j$ contributes to base model activations.

$\mathbf{d}_j^{\text{chat}}$    Decoder direction for latent $j$ in the chat model, where $\mathbf{d}_j^{\text{chat}} \in \mathbb{R}^d$. Can differ from $\mathbf{d}_j^{\text{base}}$ in both magnitude and direction.

$\widetilde{\mathbf{h}}^{\text{base}}(x)$    Reconstructed base model activation: $\widetilde{\mathbf{h}}^{\text{base}}(x) = \sum_j f_j(x)\mathbf{d}_j^{\text{base}} + \mathbf{b}^{\text{dec,base}}$.

$\widetilde{\mathbf{h}}^{\text{chat}}(x)$    Reconstructed chat model activation: $\widetilde{\mathbf{h}}^{\text{chat}}(x) = \sum_j f_j(x)\mathbf{d}_j^{\text{chat}} + \mathbf{b}^{\text{dec,chat}}$.

$\varepsilon^{\text{base}}(x)$    Reconstruction error for base model: $\varepsilon^{\text{base}}(x) = \mathbf{h}^{\text{chat}}(x) - \mathbf{h}^{\text{base}}(x)$. Captures information not explained by the crosscoder.

$\varepsilon^{\text{chat}}(x)$    Reconstruction error for chat model: $\varepsilon^{\text{chat}}(x) = \mathbf{h}^{\text{base}}(x) - \mathbf{h}^{\text{chat}}(x)$

$\Delta_{\mathbf{norm}}(j)$ Relative norm difference: $\Delta_{\text{norm}}(j) = \frac{\|\mathbf{d}_j^{\text{chat}}\|_2 - \|\mathbf{d}_j^{\text{base}}\|_2}{\max(\|\mathbf{d}_j^{\text{chat}}\|_2, \|\mathbf{d}_j^{\text{base}}\|_2)} \in [0, 1]$. Measures how chat-specific vs base-specific a latent is.

$\beta_j^{\mathbf{base}}$ Optimal scaling factor for latent $j$ to reconstruct base activations: minimizes $\sum_i \|\beta f_j(x_i)\mathbf{d}_j^{\text{chat}} - h^{\text{base}}(x_i)\|_2^2$. Intuitively, how much the chat decoder helps explain base activations.

$\beta_j^{\mathbf{chat}}$ Optimal scaling factor for latent $j$ to reconstruct chat activations (analogous to $\beta_j^{\text{base}}$)

$\nu_j$ Overall scaling ratio: $\nu_j = \beta_j^{\text{base}}/\beta_j^{\text{chat}}$. Values near 0 indicate chat-specificity; values near 1 indicate equal presence in both models.

$\nu_j^r$ Reconstruction ratio: $\nu_j^r = \beta_j^{r,\text{base}}/\beta_j^{r,\text{chat}}$, where $\beta^r$ values are computed using reconstructions instead of raw activations. Detects Latent Decoupling (high values indicate the latent's information is captured by other base latents).

$\nu_j^\varepsilon$ Error ratio: $\nu_j^\varepsilon = \beta_j^{\varepsilon,\text{base}}/\beta_j^{\varepsilon,\text{chat}}$, where $\beta^\varepsilon$ values are computed using errors. Detects Complete Shrinkage (high values indicate the latent should contribute to base reconstruction but doesn't).

$p^{\mathbf{chat}}$ Chat model's next-token probability distribution given context

$p_{\mathbf{h}^{\mathbf{chat}} \leftarrow \mathbf{h}_a}^{\mathbf{chat}}$ Modified chat model distribution when activation $h^{\text{chat}}$ is replaced with approximation $\tilde{h}$

# B Additional definitions

## B.1 L1 crosscoder

**L1 crosscoder.** Let $x$ be an string and $\mathbf{h}^{\text{base}}(x), \mathbf{h}^{\text{chat}}(x) \in \mathbb{R}^d$ denote the activations at a given layer at the last token of $x$. For a dictionary of size $D$, the latent activation of the $j^{\text{th}}$ latent $f_j(x), j \in \mathcal{J} = \{1, \dots, D\}$ is computed as

$$f_j(x) = \text{ReLU}\left(\mathbf{e}_j^{\text{base}}\mathbf{h}^{\text{base}}(x) + \mathbf{e}_j^{\text{chat}}\mathbf{h}^{\text{chat}}(x) + b_j^{\text{enc}}\right) \tag{6}$$

where $\mathbf{e}_j^{\text{base}}, \mathbf{e}_j^{\text{chat}} \in \mathbb{R}^d$ are the corresponding encoder vectors and $b_j^{\text{enc}} \in \mathbb{R}$ is the encoder bias. The reconstructed activations for both models are then defined as:

$$\widetilde{\mathbf{h}}^{\text{base}}(x) = \sum_j f_j(x)\,\mathbf{d}_j^{\text{base}} + \mathbf{b}^{\text{dec,base}} \quad \text{and} \quad \widetilde{\mathbf{h}}^{\text{chat}}(x) = \sum_j f_j(x)\,\mathbf{d}_j^{\text{chat}} + \mathbf{b}^{\text{dec,chat}} \tag{7}$$

where $\mathbf{d}_j^{\text{base}}, \mathbf{d}_j^{\text{chat}} \in \mathbb{R}^d$ are the $j^{\text{th}}$ decoder latents and $\mathbf{b}^{\text{dec,base}}, \mathbf{b}^{\text{dec,chat}} \in \mathbb{R}^d$ are the decoder biases. We define the reconstruction errors for the base and chat models as $\varepsilon^{\text{base}}(x) = \mathbf{h}^{\text{base}}(x) - \widetilde{\mathbf{h}}^{\text{base}}(x)$ and $\varepsilon^{\text{chat}}(x) = \mathbf{h}^{\text{chat}}(x) - \widetilde{\mathbf{h}}^{\text{chat}}(x)$. The training loss for the L1 crosscoder is a modified L1 SAE objective, where $\mu$ controls the sparsity weight:

$$\mathcal{L}_{\text{L1}}(x) = \frac{1}{2}\|\varepsilon^{\text{base}}(x_i)\|_2 + \frac{1}{2}\|\varepsilon^{\text{chat}}(x_i)\|_2 + \mu \sum_j f_j(x)\left(\|\mathbf{d}_j^{\text{base}}\|_2 + \|\mathbf{d}_j^{\text{chat}}\|_2\right) \tag{8}$$

While similar to training an SAE on concatenated activations, the crosscoder's sparsity loss uniquely promotes decoder norm differences (see Appendix C).

## B.2 BatchTopK crosscoder

Let $\mathcal{X} = \{x_1, \dots, x_n\}$ be a batch of $|\mathcal{X}| = n$ inputs. Following Bussmann et al. [2024], we compute the latent activation function differently during training and inference. Let $f_j(x_i)$ be the latent activation function as defined in Equation (6). Given the scaled latent activation function $v(x_i, j) = f_j(x_i)(\|\mathbf{d}_j^{\text{base}}\|_2 + \|\mathbf{d}_j^{\text{chat}}\|_2)$, the training latent activation function $f_j^{\text{train}}$ is given by:

$$f_j^{\text{train}}(x_i, \mathcal{X}) = \begin{cases} f_j(x_i) & \text{if } (x_i, j) \in \text{BATCHTOPK}(k, v, \mathcal{X}, \mathcal{J}) \\ 0 & \text{otherwise} \end{cases} \tag{9}$$

where $\text{BATCHTOPK}(k, v, \mathcal{X}, \mathcal{J})$ represents the set of indices corresponding to the top $|\mathcal{X}| \cdot k$ values of the function $v$ across all inputs $x_i \in \mathcal{X}$ and all latents $j \in \mathcal{J}$. We now redefine the reconstruction

errors and the training loss for batch $\mathcal{X}$ as follows:

$$\varepsilon^{\text{base}}(x_i, \mathcal{X}) = \mathbf{h}^{\text{base}}(x_i) - \left( \sum_j f_j^{\text{train}}(x_i, \mathcal{X})\, \mathbf{d}_j^{\text{base}} + \mathbf{b}^{\text{dec,base}} \right) \tag{10}$$

$$\varepsilon^{\text{chat}}(x_i, \mathcal{X}) = \mathbf{h}^{\text{chat}}(x_i) - \left( \sum_j f_j^{\text{train}}(x_i, \mathcal{X})\, \mathbf{d}_j^{\text{chat}} + \mathbf{b}^{\text{dec,chat}} \right) \tag{11}$$

$$\mathcal{L}_{\text{BatchTopK}}(\mathcal{X}) = \frac{1}{n} \sum_{i=1}^n \frac{1}{2}\|\varepsilon^{\text{base}}(x_i, \mathcal{X})\|_2 + \frac{1}{2}\|\varepsilon^{\text{chat}}(x_i, \mathcal{X})\|_2 + \alpha \mathcal{L}_{\text{aux}}(x_i, \mathcal{X}) \tag{12}$$

The auxiliary loss facilitates the recycling of inactive latents and is defined as $\|\varepsilon^{\text{base}}(x_i, \mathcal{X}) - \varepsilon^{\hat{\text{base}}}(x_i, \mathcal{X})\|_2 + \|\varepsilon^{\text{chat}}(x_i, \mathcal{X}) - \varepsilon^{\hat{\text{chat}}}(x_i, \mathcal{X})\|_2$, where $\varepsilon^{\hat{\text{base}}}$ and $\varepsilon^{\hat{\text{chat}}}$ represent reconstructions using only the top-$k_{\text{aux}}$ dead latents. Typically, $k_{\text{aux}}$ is set to 512 and $\alpha$ to $1/32$. For inference, we employ the following latent activation function:

$$f_j^{\text{inference}}(x_i) = \begin{cases} f_j(x_i) & \text{if } v(x_i, j) > \theta \\ 0 & \text{otherwise} \end{cases} \tag{13}$$

where $\theta$ is a threshold parameter estimated from the training data such that the number of non-zero latent activations is $k$.

$$\theta = \mathbb{E}_{\mathcal{X}} \left[ \min_{(x_i, j) \in \mathcal{X} \times \mathcal{J}} \{ v(x_i, j) \mid f_j^{\text{train}}(x_i, \mathcal{X}) > 0 \} \right] \tag{14}$$

## B.3 Alternative BatchTopK variations

We experimented with several variations of the BatchTopK activation function to investigate whether alternative sparsity mechanisms could further improve the identification of *chat-specific* latents. However, none of these variations yielded more *chat-specific* latents than the BatchTopK approach described above, so we focus on this version in the main paper.

**Concatenated decoder norm variant.** The first variation modifies the scaling function $v(x_i, j)$ used in the top-$k$ selection. Instead of summing the decoder norms as in our approach, we use the norm of the concatenated decoder vectors:

$$v'(x_i, j) = f_j(x_i)\|[\mathbf{d}_j^{\text{base}}, \mathbf{d}_j^{\text{chat}}]\|_2 \tag{15}$$

where $[\mathbf{d}_j^{\text{base}}, \mathbf{d}_j^{\text{chat}}] \in \mathbb{R}^{2d}$ denotes the concatenation of both decoder vectors. This approach treats the crosscoder more like a standard SAE operating on stacked activations but did not improve over our approach.

**Model-independent BatchTopK variant.** The second variation computes BatchTopK selection independently for each model, using the model-specific scaling function

$$v^M(x_i, j) = f_j(x_i)\|\mathbf{d}_j^M\|_2 \tag{16}$$

for model $M \in \{\text{base}, \text{chat}\}$. This approach was motivated by the observation that standard Batch-TopK has an inherent bias toward shared latents. Since latents are selected based on their total reconstruction benefit across both models, a shared latent that reduces loss by 0.6 on each model (total benefit 1.2) will be preferred over a model-specific latent that reduces loss by 1.0 on one model and 0 on the other (total benefit 1.0). We hypothesized that this bias might prevent discovery of important chat-specific features introduced during fine-tuning, as they would be crowded out by shared representations. The model-independent variant removes this bias by allowing each model to allocate its $k$ budget independently, potentially revealing chat-specific latents that would otherwise be suppressed. As expected, the model-independent variant produced more *chat-only* latents. However, these additional latents suffered from increased latent decoupling issues, ultimately not yielding more *chat-specific* latents by our $\nu^r$ and $\nu^\varepsilon$ metrics. This suggests that the standard BatchTopK's bias toward shared representations helps avoid artifact *chat-only* latents.

## C  Comparing sparsity losses: Crosscoder vs. stacked SAE

An L1 crosscoder can be viewed as an SAE operating on stacked activations, where the encoder and decoder vectors are similarly stacked:

$$\mathbf{h}(x) = \left[\mathbf{h}^{\text{base}}(x), \quad \mathbf{h}^{\text{chat}}(x)\right] \in \mathbb{R}^{2d} \tag{17}$$

$$\mathbf{e}_j = \left[\mathbf{e}_j^{\text{base}}, \quad \mathbf{e}_j^{\text{chat}}\right] \in \mathbb{R}^{2d} \tag{18}$$

$$\mathbf{d}_j = \left[\mathbf{d}_j^{\text{base}}, \quad \mathbf{d}_j^{\text{chat}}\right] \in \mathbb{R}^{2d} \tag{19}$$

$$\mathbf{b}^{\text{dec}} = \left[\mathbf{b}^{\text{dec,base}}, \mathbf{b}^{\text{dec,chat}}\right] \tag{20}$$

The reconstruction remains equivalent because

$$f_j(x) = \text{ReLU}\left(\mathbf{e}_j\,\mathbf{h} + b_j^{\text{enc}}\right) \tag{21}$$

$$= \text{ReLU}\left(\mathbf{e}_j^{\text{base}}\mathbf{h}^{\text{base}}(x) + \right.$$
$$\left. \mathbf{e}_j^{\text{chat}}\mathbf{h}^{\text{chat}}(x) + b_j^{\text{enc}}\right) \tag{22}$$

and hence,

$$\left[\mathbf{h}^{\tilde{\text{base}}}(x), \quad \mathbf{h}^{\tilde{\text{chat}}}(x)\right] = \sum_j f_j(x)\mathbf{d}_j + \mathbf{b}^{\text{dec}} \tag{23}$$

However, the key difference arises in the sparsity loss. For the crosscoder, the sparsity loss is given by:

$$L_{\text{sparsity}}^{\text{crosscoder}}(x) = \sum_j f_j(x)\left(\sqrt{\sum_{i=1}^{d}(\mathbf{d}_{j,i}^{\text{chat}})^2} + \sqrt{\sum_{i=1}^{d}(\mathbf{d}_{j,i}^{\text{base}})^2}\right) \tag{24}$$

For a stacked SAE, it is:

$$L_{\text{sparsity}}^{\text{SAE}}(x) = \sum_j f_j(x)\sqrt{\sum_{i=1}^{2d}(\mathbf{d}_{j,i})^2}$$

$$= \sum_j f_j(x)\sqrt{\sum_{i=1}^{d}(\mathbf{d}_{j,i}^{\text{base}})^2 + \sum_{i=1}^{d}(\mathbf{d}_{j,i}^{\text{chat}})^2} \tag{25}$$

The difference between $\sqrt{x+y}$ and $\sqrt{x} + \sqrt{y}$ introduces an inductive bias in the crosscoder that encourages the norm of one decoder (often the base decoder) to approach zero when the corresponding latent is only informative in one model.

Figure 9 displays a heatmap of the functions $\sqrt{x^2 + y^2}$ and $\sqrt{x^2} + \sqrt{y^2}$ along with their negative gradients, as visualized by the arrows. One can observe that for the crosscoder sparsity variant $\sqrt{x^2} + \sqrt{y^2}$ the gradient encourages the norm of one of the decoders to approach zero much more quickly compared to the SAE's $\sqrt{x^2 + y^2}$.

## D  Illustrative example of Latent Decoupling

As a reminder, Latent Decoupling happens when a *chat-only* latent $j$ is also present in the base activations but is reconstructed by other base decoder latents. To spell it out in more details, consider the following set up: a concept C may be represented identically in both models by some direction $\mathbf{d}_C$ but activate on different non-exclusive data subsets. Let $f_C^{\text{chat}}(x)$ and $f_C^{\text{base}}(x)$ be concept C's optimal activation functions in chat and base models, defined as $f_C^{\text{chat}}(x) = f_{\text{shared}}(x) + f_{\text{c-excl}}(x)$ and $f_C^{\text{base}}(x) = f_{\text{shared}}(x) + f_{\text{b-excl}}(x)$, where $f_{\text{shared}}$ encodes shared activation, while $f_{\text{b-excl}}$ and $f_{\text{c-excl}}$ define model exclusive activations. For interpretability, the crosscoder should ideally learn three latents:

1. A *shared* latent $j_{\text{shared}}$ representing C when active in both models using $f_{j_{\text{shared}}} = f_{\text{shared}}$ and $\mathbf{d}_{\text{chat}} = \mathbf{d}_{\text{base}} = \mathbf{d}_C$,

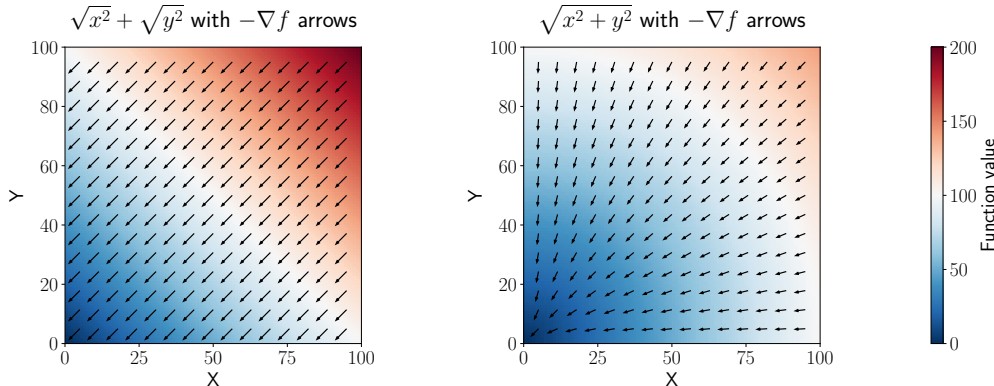

Figure 9: Heatmap comparing the two functions $\sqrt{x^2 + y^2}$ and $\sqrt{x^2} + \sqrt{y^2}$ along with their negative gradients.

2. A *chat-only* latent $j_{\text{chat}}$ representing C when exclusively active in the chat model using $f_{j_{\text{chat}}} = f_{\text{c-excl}}$ and $\mathbf{d}_{\text{chat}} = \mathbf{d}_C, \mathbf{d}_{\text{base}} = \mathbf{0}$, and
3. A *base-only* latent $j_{\text{base}}$ representing C when exclusively active in the base model using $f_{j_{\text{base}}} = f_{\text{b-excl}}$ and $\mathbf{d}_{\text{chat}} = \mathbf{0}, \mathbf{d}_{\text{base}} = \mathbf{d}_C$.

However, the L1 crosscoder achieves equivalent loss using just two latents:

1. A *chat-only* latent $j_{\text{chat}}$ representing C in the chat model using $f_{j_{\text{chat}}} = f_{\text{c-excl}} + f_{\text{shared}}$ and $\mathbf{d}_{\text{chat}} = \mathbf{d}_C, \mathbf{d}_{\text{base}} = \mathbf{0}$, and
2. A *base-only* latent $j_{\text{base}}$ representing C in the base model using $f_{j_{\text{base}}} = f_{\text{b-excl}} + f_{\text{shared}}$ and $\mathbf{d}_{\text{chat}} = \mathbf{0}, \mathbf{d}_{\text{base}} = \mathbf{d}_C$. In this scenario, the so-called "*chat-only*" latent is only truly chat-only on a subset of its activation pattern.

Although whenever $f_{\text{shared}} > 0$ two latents are active instead of one, the sparsity loss is the same because the sparsity loss includes the decoder vector norms. [12] To illustrate the phenomenon of Latent Decoupling we choose the oversimplified case where $f_{\text{b-excl}}(x) = f_{\text{c-excl}}(x) = 0$. Let us consider a latent $j$ with $f_j(x) = \alpha$. On the other hand, let there be two other latents $p$ and $q$ with

$$\mathbf{d}_p^{\text{base}} = \mathbf{d}_j^{\text{base}}, \quad \mathbf{d}_p^{\text{chat}} = \mathbf{0}$$
$$\mathbf{d}_q^{\text{base}} = \mathbf{0}, \quad \mathbf{d}_q^{\text{chat}} = \mathbf{d}_j^{\text{chat}}$$

and $f_p(x) = f_q(x) = \alpha$. Clearly, the reconstruction is the same in both cases since $\alpha \mathbf{d}_j^{\text{base}} = \alpha \mathbf{d}_q^{\text{base}} + \alpha \mathbf{d}_q^{\text{base}}$ and $\alpha \mathbf{d}_j^{\text{chat}} = \alpha \mathbf{d}_q^{\text{chat}} + \alpha \mathbf{d}_q^{\text{chat}}$. Further, the L1 regularization term is the same since

$$\alpha \left( ||\mathbf{d}_j^{\text{base}}||_2 + ||\mathbf{d}_j^{\text{chat}}||_2 \right) = \tag{26}$$
$$\alpha \left( ||\mathbf{d}_p^{\text{base}}||_2 + ||\mathbf{d}_p^{\text{chat}}||_2 \right)$$
$$+ \alpha \left( ||\mathbf{d}_q^{\text{base}}||_2 + ||\mathbf{d}_q^{\text{chat}}||_2 \right)$$
$$= \alpha \left( ||\mathbf{d}_p^{\text{base}}||_2 + 0 \right) + \alpha \left( 0 + ||\mathbf{d}_q^{\text{chat}}||_2 \right) \tag{27}$$

Hence both solutions achieve the exact same loss under the L1 crosscoder.

However, the BatchTopK crosscoder actively encourages the three-latent solution. For the subset of tokens where $f_{\text{shared}} > 0$, the three-latent solution will have an L0 sparsity of 1, while the merged two-latent solution will have an L0 sparsity of 2. Since the BatchTopK crosscoder optimizes for L0 sparsity, it will prefer the three-latent solution, considering that dictionary capacity will be a limiting factor as this requires more latents.

---

[12]In the simplest case where $f_{\text{c-excl}}(x) = f_{\text{b-excl}}(x) = 0$, there exists a *base-only* latent $j_{\text{twin}}$ with $\mathbf{d}_j^{\text{chat}} = \mathbf{d}_{j_{\text{twin}}}^{\text{base}}$ and identical activation function that reconstructs the information of $\mathbf{d}_j^{\text{chat}}$ in the base model. The sparsity loss equals that of a single shared latent.

# E  More details regarding Latent Scaling

## E.1  Closed form solution for Latent Scaling

Consider a latent $j$ with decoder vector $\mathbf{d}$. Our goal is to find the optimal scaling factor $\beta$ that minimizes the squared reconstruction error:

$$\underset{\beta}{\arg\min} \sum_{i=0}^{n} \|\beta f(x_i)\mathbf{d} - \mathbf{y}\|_2^2 \tag{28}$$

To solve this optimization problem efficiently, we reformulate it in matrix form. Let $\mathbf{Y} \in \mathbb{R}^{n \times d}$ be the stacked data matrix and $\mathbf{f} \in \mathbb{R}^n$ be the vector of latent activations for latent $j$ across all datapoints. The objective can then be expressed using the Frobenius norm of the residual matrix $\mathbf{R} = \beta \mathbf{f}\mathbf{d}^T - \mathbf{Y}$, where $\mathbf{f}\mathbf{d}^T \in \mathbb{R}^{n \times d}$ represents the outer product of the latent activation vector and decoder vector. Our minimization problem becomes:

$$
\begin{aligned}
\|\mathbf{R}\|_F^2 &= \|\beta \mathbf{f}\mathbf{d}^T - \mathbf{Y}\|_F^2 \\
&= \mathrm{Tr}\left[(\beta \mathbf{f}\mathbf{d}^T - \mathbf{Y})^\top (\beta \mathbf{f}\mathbf{d}^T - \mathbf{Y})\right] \\
&= \mathrm{Tr}\left[\mathbf{Y}^\top \mathbf{Y}\right] - 2\beta \mathrm{Tr}\left[\mathbf{Y}^\top \mathbf{f}\mathbf{d}^T\right] \\
&\quad + \beta^2 \mathrm{Tr}\left[(\mathbf{f}\mathbf{d}^T)^\top \mathbf{f}\mathbf{d}^T\right]
\end{aligned}
$$

Using trace properties, we get:

$$\mathrm{Tr}\left[\mathbf{Y}^\top \mathbf{f}\mathbf{d}^T\right] = \mathbf{d}^\top (\mathbf{Y}^\top \mathbf{f})$$

$$\mathrm{Tr}\left[(\mathbf{f}\mathbf{d}^T)^\top \mathbf{f}\mathbf{d}^T\right] = \|\mathbf{f}\|_2^2 \|\mathbf{d}\|_2^2$$

Taking the derivative with respect to $\beta$ and setting it to zero:

$$\frac{\delta}{\delta \beta} \|\mathbf{R}\|_F^2 = -2\mathbf{d}^\top (\mathbf{Y}^\top \mathbf{f}) + 2\beta \|\mathbf{f}\|_2^2 \|\mathbf{d}\|_2^2 = 0$$

This yields the closed form solution:

$$\beta = \frac{\mathbf{d}^\top (\mathbf{Y}^\top \mathbf{f})}{\|\mathbf{f}\|_2^2 \|\mathbf{d}\|_2^2} = \frac{\langle \mathbf{Y}\mathbf{d}, \mathbf{f} \rangle}{\|\mathbf{f}\|_2^2 \|\mathbf{d}\|_2^2} \tag{29}$$

Without loss of generality, we can assume $\mathbf{d}$ has unit norm.[13]

To gain intuition for this formula, consider a simplified toy setting where $f_i \in \{0, 1\}$ (latent either fires or doesn't) and $(\mathbf{Y}\mathbf{d})_i \in \{0, \alpha\}$ (the target contains the concept with magnitude $\alpha$ or not at all). In this case, the closed form simplifies to:

$$\beta = \frac{\sum_i (\mathbf{Y}\mathbf{d})_i f_i}{\sum_i f_i^2} \tag{30}$$

$$= \alpha \frac{\#\{i : f_i \neq 0 \text{ and } (\mathbf{Y}\mathbf{d})_i \neq 0\}}{\#\{i : f_i \neq 0\}} \tag{31}$$

$$= \alpha \cdot P(\text{concept present in target} \mid \text{latent active}) \tag{32}$$

This toy example illustrates that $\beta$ captures both the magnitude $\alpha$ at which the concept appears in the target activations and the conditional probability that the concept is actually present when the latent fires. For a truly fine-tuning-specific latent, we expect this conditional probability to be near 0 for the base model activations (yielding $\beta \approx 0$) and near 1 for the fine-tuned model activations (yielding $\beta \approx \alpha$). In contrast, a shared latent should exhibit similar $\beta$ values across both model activations, reflecting consistent presence of the underlying concept.

---

[13]By defining $f' = \|\mathbf{d}\|_2 f$ and $\mathbf{d}' = \mathbf{d}/\|\mathbf{d}\|_2$, we obtain an equivalent formulation with unit decoder norm.

## E.2 Detailed setup for Latent Scaling

We specify the exact target vectors $\mathbf{y}$ used in Equation (28) for computing the different $\beta$ values to compute our chat-specificity metrics. To measure how well latent $j$ explains the reconstruction *error*, we exclude latent $j$ from the reconstruction. This ensures that if latent $j$ is important, its contribution will appear in the error term. For chat-only latents, we expect distinct behavior in each model: no contribution in the base model ($\beta_j^{\varepsilon,\text{base}} \approx 0$) but strong contribution in the chat model ($\beta_j^{\varepsilon,\text{chat}} \approx 1$), resulting in $\nu_j^{\varepsilon} \approx 0$. In contrast, *shared* latents should have similar contributions in both models, resulting in approximately equal values for $\beta_j^{\varepsilon,\text{base}}$ and $\beta_j^{\varepsilon,\text{chat}}$ and consequently $\nu_j^{\varepsilon} \approx 1$.

$$\beta_j^{\varepsilon,\text{base}} : \mathbf{y}_i = \mathbf{h}^{\text{base}}(x_i) - \sum_{k,k \neq j} f_k(x_i)\, \mathbf{d}_k^{\text{base}} + \mathbf{b}^{\text{dec,base}} \tag{33}$$

$$\beta_j^{\varepsilon,\text{chat}} : \mathbf{y}_i = \mathbf{h}^{\text{chat}}(x_i) - \sum_{k,k \neq j} f_k(x_i)\, \mathbf{d}_k^{\text{chat}} + \mathbf{b}^{\text{dec,chat}} \tag{34}$$

To measure how well a latent $j$ explains the *reconstruction*, we simply use

$$\beta_j^{r,\text{base}} : \quad \mathbf{y}_i = \widetilde{\mathbf{h}}^{\text{base}}(x_i) \tag{35}$$

$$\beta_j^{r,\text{chat}} : \quad \mathbf{y}_i = \widetilde{\mathbf{h}}^{\text{chat}}(x_i) \tag{36}$$

In a similar manner, we expect the fraction $\nu_j^r$ to be low for chat-only latents and around 1 for *shared* latents. For all of our analyses, we filter out latents with negative $\beta^{\text{base}}$ values (L1: 46 in reconstruction and 1 in error, None in BatchTopK ). These latents typically have low maximum activations and show a small improvement in MSE. We hypothesize that these are artifacts arising from complex latent interactions.

## E.3 Additional analysis for Latent Scaling

Figure 10a and Figure 10b analyze the relationship between our scaling metrics ($\nu^{\varepsilon}$ and $\nu^r$) and the actual improvement in reconstruction quality in the L1 crosscoder. For each latent, we compute the MSE improvement as:

$$\text{MSEImprovement} = \frac{\text{MSE}_{\text{original}} - \text{MSE}_{\text{scaled}}}{\text{MSE}_{\text{original}}}$$

where $\text{MSE}_{\text{scaled}}$ is measured after applying our Latent Scaling technique. We then examine the ratio of MSE improvements between the base and chat models, analogous to our $\nu$ metrics. The strong correlation between the $\nu$ values and MSE improvement ratios validates that our scaling approach captures meaningful differences in how latents contribute to reconstruction in each model.

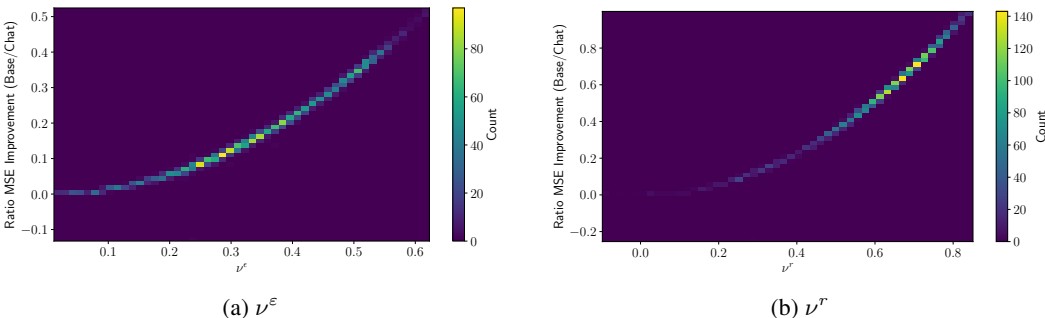

(a) $\nu^{\varepsilon}$          (b) $\nu^r$

Figure 10: Comparison of the ratio of MSE improvement compared to the value of $\nu^{\varepsilon}$ and $\nu^r$.

In Figure 11, we analyze the Latent Scaling technique by examining its relationship with the $\Delta_{\text{norm}}$ score. Specifically, we identify the 100 latents with the lowest $\nu^{\varepsilon}$ values and analyze their rankings according to the $\Delta_{\text{norm}}$ metric. As shown in Figure 11, there is limited correlation between the two measures - simply using a lower NormDiff threshold to identify *chat-only* latents produces substantially different results from our Latent Scaling approach.

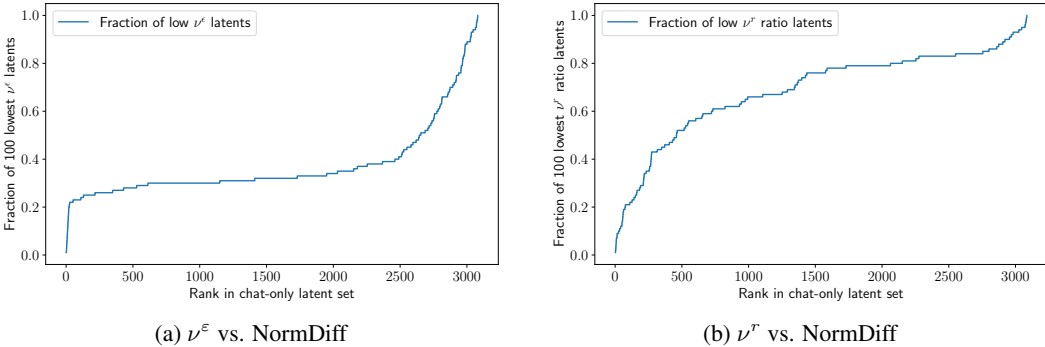

(a) $\nu^\varepsilon$ vs. NormDiff

(b) $\nu^r$ vs. NormDiff

Figure 11: Comparison of latent rankings between $\nu$ and NormDiff scores. The lines shows the fraction of the 100 latents with the lowest $\nu$ values ($x$-axis) that have a rank lower than the given rank under the NormDiff score ($y$-axis).

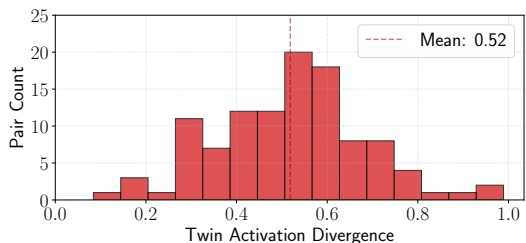

Figure 12: Distribution activation divergence over high cosine similarity (*chat-only*, *base-only*) latent pairs. 1 means that latents never have high activations ($> 0.7 \times$ `max_activation`) at the same time, 0 means that high activations correlate perfectly.

# F   Cosine similarity of coupled latents.

As further evidence for Latent Decoupling occuring, we compute the cosine similarity between $\{\mathbf{d}_j^{\text{chat}}, j \in \textit{chat-only}\}$ and $\{\mathbf{d}_j^{\text{base}}, j \in \textit{base-only}\}$ revealing 109 $(j, j_{\text{twin}})$ pairs where $\text{cosim}(\mathbf{d}_j^{\text{chat}}, \mathbf{d}_{j_{\text{twin}}}^{\text{base}}) > 0.9$. To quantify activation pattern overlap between twins $(j, j_{\text{twin}})$, we introduce an *activation divergence score* from 0 (always co-activate) to 1 (never co-activate) (see Appendix F.1). Figure 12 shows the divergence distribution across these pairs, highlighting that 60% of the pairs primarily activate on different contexts, with some pairs almost exclusively firing on different contexts (divergence of 1), while others exhibit substantial overlapping activations. This analysis demonstrates two important insights:

1. The Latent Decoupling phenomenon described in Appendix D, where the crosscoder learns a *base-only* and a *chat-only* latent that partially activate together instead of learning a *shared* latent, is empirically observed in practice.
2. Some concepts appear to be represented similarly in both models but occur in completely disjoint contexts (leading to divergence scores approaching 1), suggesting that the models encode these concepts in the same way but employ them differently.

Additionally, we find no pairs of *chat-only* latents and $\Delta_{\text{norm}} < 0.6$ latents with a cosine similarity greater than 0.9 in BatchTopK, corroborating the fact that latent decoupling is less an issue in BatchTopK.

### F.1 Detailed setup for activation divergence

In order to compute the activation divergence we compute for each pairs $p = (i, j)$, we first compute the max pair activation $A_p$ on the training set $D_{\text{train}}$ (containing data from LMSYS and FineWeb)

$$A_p = \max(A_i, A_j)$$
$$A_i = \max\{f_i(x)(\|\mathbf{d}_i^{\text{chat}}\| + \|\mathbf{d}_i^{\text{base}}\|), x \in D_{\text{train}}\}$$

Then the divergence $\text{Div}_p$ is computed as follow

$$\text{Div}_p = \frac{\text{Single}_p}{\text{High}_p}$$

$$\text{Single}_p = \#\text{single}_i + \#\text{single}_j$$

$$\text{High}_p = \#(\text{high}_i \cup \text{high}_j)$$

where $\#\text{single}_i$ is the set of input $x \in D_{\text{val}}$ where $i$ has a high activation but not $j$ and $\text{high}_i$ is the total number of high activations computed as follows:

$$\text{only}_i = \{x \in D_{\text{val}}, f_i(x) > 0.7A_p \wedge f_j(x) < 0.3A_p\}$$
$$\text{high}_i = \{x \in D_{\text{val}}, f_i(x) > 0.7A_p\}$$

## G   Causality experiments

### G.1   Reproduction on LMSYS-CHAT

In Figure 13 we repeat the causality experiments from Section 3.2 for the L1 crosscoder on 700'000 tokens from the LMSYS-CHAT dataset, that the crosscoder was trained on. Note that while this dataset is much larger, the model responses are not generated by the Gemma 2 2b it model, and hence the model answers are out of distribution for this model. Since this dataset is much larger, the confidence intervals are much smaller. The results are qualitatively similar to the ones on the generated dataset in the main paper.

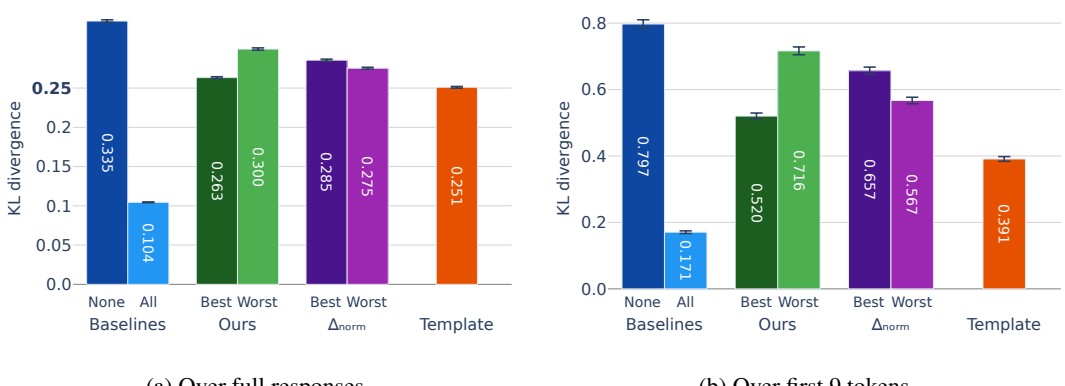

(a) Over full responses.                    (b) Over first 9 tokens.

Figure 13: Comparison of KL divergence between different approximations of chat model activations on the LMSYS-CHAT dataset. We establish baselines by replacing either *None* or *All* of the latents. We then evaluate our Latent Scaling metric (*Ours*) against the relative norm difference ($\Delta_{\text{norm}}$) by comparing the effects of replacing the top and bottom 50% of latents ranked by each metric (*Best* vs *Worst*). Additionally, we measure the impact of replacing activations only on template tokens (*Template*). We show the 95% confidence intervals for all measurements. Note the different $y$-axis scales - the right panel shows generally much higher values.

## H   Autointerpretability details

We automatically interpret the identified latents using the pipeline from Paulo et al. [2024]. To explain the latents, we provide ten activating examples from each activation tercile to Llama 3.3

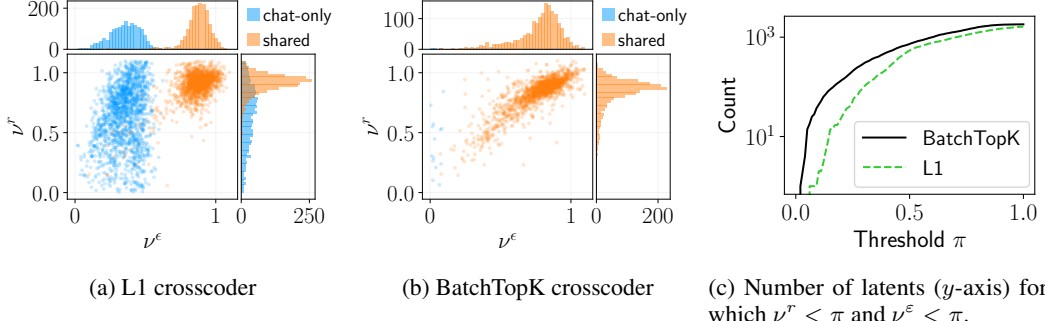

(a) L1 crosscoder      (b) BatchTopK crosscoder      (c) Number of latents ($y$-axis) for which $\nu^r < \pi$ and $\nu^\varepsilon < \pi$.

Figure 14: We compare how **Llama3.2 1B** *chat-only* latents are affected by the issues described in Section 2.2. Left/Middle: $\nu$ distributions for L1 and BatchTopK crosscoders, with each point representing a single latent. High $\nu^r$ values ($y$-axis) overlapping with *shared* distribution indicate Latent Decoupling (redundant encoding). High $\nu^\varepsilon$ values ($x$-axis) shows Complete Shrinkage (useful base latents forced to zero norm). Low values on both metrics identify truly chat-specific latents. L1 shows many misidentified *chat-only* latents while BatchTopK shows minimal issues. Right: Count of latents below a range of $\nu$ thresholds ($x$-axis), comparing 1844 L1 *chat-only* latents versus top-1844 BatchTopK latents sorted by $\Delta_{\text{norm}}$.

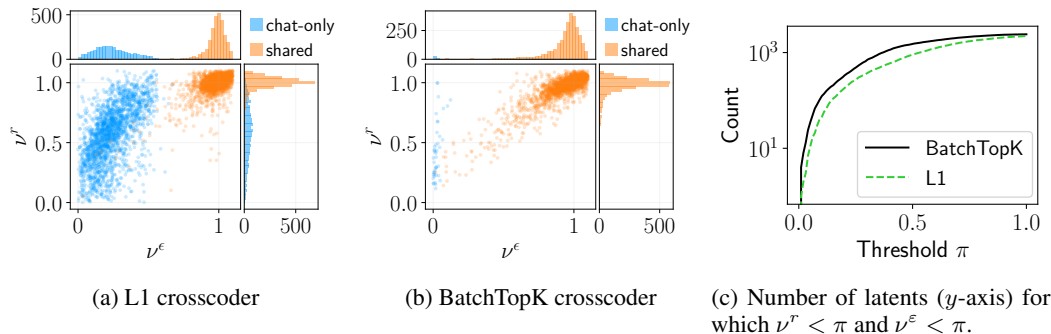

(a) L1 crosscoder      (b) BatchTopK crosscoder      (c) Number of latents ($y$-axis) for which $\nu^r < \pi$ and $\nu^\varepsilon < \pi$.

Figure 15: We compare how **Llama3.1 8B** *chat-only* latents are affected by the issues described in Section 2.2. Left/Middle: $\nu$ distributions for L1 and BatchTopK crosscoders, with each point representing a single latent. High $\nu^r$ values ($y$-axis) overlapping with *shared* distribution indicate Latent Decoupling (redundant encoding). High $\nu^\varepsilon$ values ($x$-axis) shows Complete Shrinkage (useful base latents forced to zero norm). Low values on both metrics identify truly chat-specific latents. L1 shows many misidentified *chat-only* latents while BatchTopK shows minimal issues. Right: Count of latents below a range of $\nu$ thresholds ($x$-axis), comparing 2442 L1 *chat-only* latents versus top-2442 BatchTopK latents sorted by $\Delta_{\text{norm}}$.

70B [Grattafiori et al., 2024]. Latents are scored using a modified detection metric from Paulo et al. [2024]. We provide ten new activating examples from each tercile. Rather than comparing activation examples against randomly selected non-activating examples, we use semantically similar non-activating examples identified through Sentence BERT embedding similarity [Reimers and Gurevych, 2019] using the *all-MiniLM-L6-v2* model. To find these similar examples, we join all activating examples into a single string and embed it, then compute similarity scores against embeddings for each window of tokens to identify the most semantically related non-activating examples. This is a strictly harder task than scoring activation examples against a random set of non-activating examples.

# I Reproducing results on other models

## I.1 Llama models

We reproduce our experiments on both *Llama3.2 1B* and *Llama3.1 8B* models [Grattafiori et al., 2024]. Different from the Gemma models, the Llama models have a very different embedding for some of the template tokens. We replace several template tokens with single token alternatives:

- `<start_header_id>` is replaced with `\n\n\n`
- `<eot_id>` is replaced with `####`
- `<end_header_id>` is replaced with `####`

For Llama3.2 1B, we use the same training pipeline as the main paper with $\mu = 3.6e - 2$ for the L1 crosscoder, resulting in an L0 of 110 after training. We compare this to a BatchTopK crosscoder with $k = 100$. While this k value differs slightly, retraining would be computationally expensive, and the lower k actually disadvantages the BatchTopK crosscoder. The L1 crosscoder achieves 76.5% validation FVE while the BatchTopK crosscoder achieves 81.5%.

For Llama3.1 8B, we use $\mu = 2.1e - 2$ for the L1 crosscoder, resulting in an L0 of 201, compared against a BatchTopK crosscoder with $k = 200$. For the BatchTopK crosscoder, we make two key modifications compared to the other models: 1) we initialize the encoder and decoder norms to 0.3 instead of 1.0 which is crucial for convergence, and 2) we anneal $k$ from 1000 to 200 over 5000 steps to prevent dead latents. The L1 crosscoder achieves 76.6% validation FVE while the BatchTopK crosscoder achieves 81.5%. Due to computational constraints, we only use 10M tokens to train the latent scalers $\beta$.

Both models exhibit consistent patterns. The L1 crosscoders systematically overidentify *chat-only* latents:

- For Llama3.2 1B (Figure 14), the $\nu$ distributions reveal numerous misidentified *chat-only* latents in the L1 crosscoder, while the BatchTopK shows minimal issues. In Figure 14c we see that the BatchTopK crosscoder effectively identifies more truly chat-specific latents.
- The same patterns hold for Llama3.1 8B, as shown in Figure 15.

## I.2 Reproducing on chat model fine-tuned on narrower domains

To verify that our findings extend beyond the base vs. chat phenomenon, we conducted additional experiments on models fine-tuned in narrower domains. We compare two domain-specific fine-tuning scenarios:

- **Medical domain fine-tuning:** We compare `google/gemma-2-2b-it` to `OpenMeditron/Meditron3-Gemma2-2B` from the Meditron3 Sallinen et al. [2025] suite. Crosscoders were trained on 50M tokens from LMSYS and 39M tokens of medical data, including a mixture of [Tran et al., 2024, `bio-nlp-umass/bioinstruct`], [Chen et al., 2024, `FreedomIntelligence/medical-o1-reasoning-SFT`], and [Xiong et al., 2024, `MedRAG/pubmed`].
- **RL fine-tuning on reasoning data:** We compare `deepseek-ai/DeepSeek-R1-Distill-Qwen-1.5B` to `nvidia/Nemotron-Research-Reasoning-Qwen-1.5B`, which applies extended RL training periods for deeper exploration of reasoning strategies Liu et al. [2025]. Crosscoders were trained on 50M tokens from LMSYS and 50M tokens of reasoning traces from `open-r1/OpenR1-Math-220k`.

For both comparisons, we trained L1 and BatchTopK crosscoders with comparable $L_0 \approx 100$ on the validation set and measured how many latents are truly specific to the fine-tuned model as determined by Latent Scaling. Table 1 shows results across all investigated models, including the number of fine-tuned-only (FT-only) latents based on the relative norm difference $\Delta$.

Figure 16 shows the medical domain fine-tuning results, demonstrating the same systematic patterns observed in base vs. chat comparisons. The L1 crosscoder identifies 246 fine-tuning-only latents with $\Delta \geq 0.9$, but 235 of these (95.5%) exhibit high reconstruction ratios $\nu > 0.6$, indicating false

Table 1: Domain-specific fine-tuning results across different model pairs, architectures, and fine-tuning methods. The table shows the systematic pattern where L1 crosscoders consistently misidentify shared latents as fine-tuning-only due to Complete Shrinkage and Latent Decoupling phenomena.

| Model | Type | # FT-only ($\Delta \geq 0.9$) | False FT-only ($\nu > 0.6$) | # latents $< \pi$ | | | |
| --- | --- | --- | --- | --- | --- | --- | --- |
| | | | | **0.2** | **0.4** | **0.6** | **0.8** |
| Gemma2-2B-Chat | BatchTopK | 134 | 1 (0.7%) | 301 | 979 | 2035 | 3269 |
| | L1 | 3176 | 2132 (67.1%) | 13 | 201 | 982 | 2970 |
| Llama-3.1-8B-Chat | BatchTopK | 97 | 13 (13.4%) | 382 | 1263 | 2073 | 2848 |
| | L1 | 2442 | 1210 (49.5%) | 234 | 765 | 1594 | 2440 |
| Llama-3.2-1B-Chat | BatchTopK | 17 | 2 (11.8%) | 137 | 517 | 1109 | 1990 |
| | L1 | 1844 | 1071 (58.1%) | 24 | 236 | 790 | 1330 |
| Qwen-1.5B-Nemotron | BatchTopK | 0 | 0 (0.0%) | 0 | 2 | 22 | 127 |
| | L1 | 59 | 58 (98.3%) | 0 | 0 | 2 | 24 |
| Meditron3-Gemma | BatchTopK | 0 | 0 (0.0%) | 13 | 55 | 158 | 529 |
| | L1 | 246 | 235 (95.5%) | 7 | 21 | 35 | 204 |

attribution due to Complete Shrinkage or Latent Decoupling. In contrast, the BatchTopK crosscoder identifies 0 false fine-tuning-only latents (0.0%).

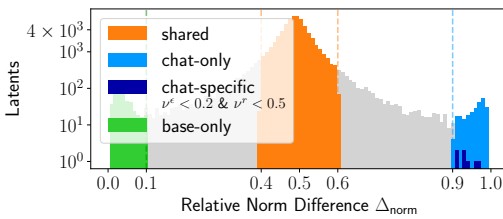

(a) L1 decoder norm differences for medical domain fine-tuning (Gemma-2-2b-it vs. Meditron3).

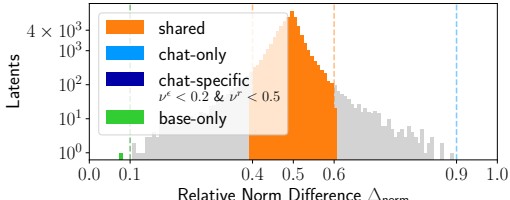

(b) BatchTopK decoder norm differences for medical domain fine-tuning (Gemma-2-2b-it vs. Meditron3).

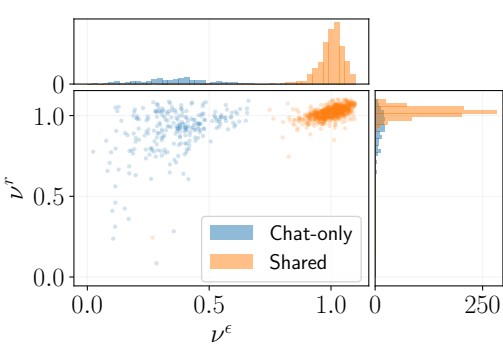

(c) L1 error vs reconstruction ratio for medical domain fine-tuning, showing Complete Shrinkage and Latent Decoupling patterns.

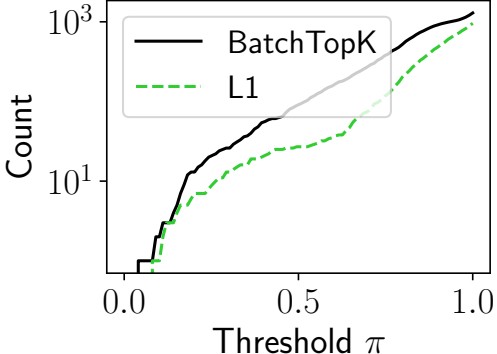

(d) Latents vs threshold comparison for medical domain fine-tuning, comparing L1 and BatchTopK identification of domain-specific latents.

Figure 16: Domain-specific fine-tuning results for medical domain (Gemma-2-2b-it vs. Meditron3-Gemma2-2B). **Top:** Decoder norm differences for L1 (left) and BatchTopK (right) crosscoders. **Bottom:** L1 error vs reconstruction analysis (left) and threshold comparison (right). The results demonstrate that L1 crosscoders systematically misidentify shared medical concepts as fine-tuning-only, while BatchTopK crosscoders more accurately identify genuinely domain-specific latents. Medical fine-tuning was performed on 39M tokens of medical data including bioinstruct, medical reasoning, and PubMed content.

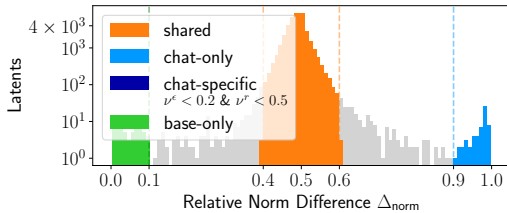
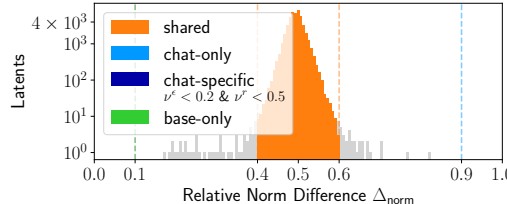

(a) L1 decoder norm differences for reasoning domain fine-tuning (R1dist-Qwen-1.5B vs. Nemotron).

(b) BatchTopK decoder norm differences for reasoning domain fine-tuning (R1dist-Qwen-1.5B vs. Nemotron).

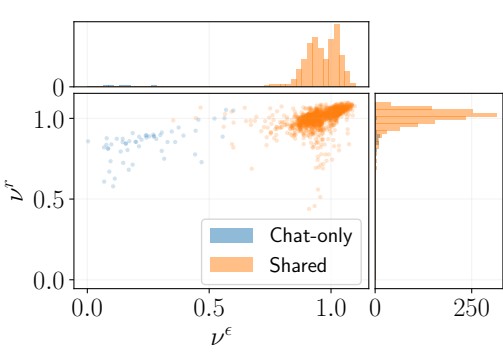
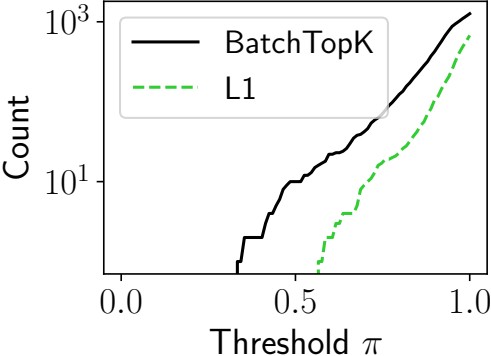

(c) L1 error vs reconstruction ratio for reasoning domain fine-tuning, showing Complete Shrinkage and Latent Decoupling patterns.

(d) Latents vs threshold comparison for reasoning domain fine-tuning, comparing L1 and BatchTopK identification of domain-specific latents.

Figure 17: Domain-specific fine-tuning results for reasoning domain (DeepSeek-R1-Distill-Qwen-1.5B vs. Nemotron-Research-Reasoning-Qwen-1.5B). **Top:** Decoder norm differences for L1 (left) and BatchTopK (right) crosscoders. **Bottom:** L1 error vs reconstruction analysis (left) and threshold comparison (right). The reasoning domain shows the most extreme misattribution patterns, with 98.3% of L1-identified latents being false positives. RL fine-tuning was performed on 50M tokens of reasoning traces from OpenR1-Math-220k.

The reasoning domain comparison (Figure 17) shows even more extreme patterns. For the DeepSeek-R1 vs. Nemotron-Reasoning comparison (Qwen-1.5B-Nemotron), the L1 crosscoder identifies 59 reasoning-related latents as fine-tuning-only with $\Delta \geq 0.9$, but 58 of these (98.3%) exhibit Complete Shrinkage or Latent Decoupling with $\nu > 0.6$ - the highest false attribution rate across all model pairs. The BatchTopK crosscoder again identifies 0 false fine-tuning-only latents (0.0%).

We observe two consistent patterns across all models in Table 1: (i) The $\Delta$ metric in L1 crosscoders consistently identifies a large number of latents as fine-tuning-only that actually display Complete Shrinkage or Latent Decoupling, with false attribution rates ranging from 49.5% to 98.3%. (ii) BatchTopK crosscoders maintain low false attribution rates (0.0% to 13.4%) and consistently identify more genuinely fine-tuning-specific latents when using Latent Scaling.

These results demonstrate that our findings reproduce across narrow domain fine-tuning (medical & reasoning), different architectures (Qwen & Llama), and alternative fine-tuning algorithms (RL tuning), supporting the generality and robustness of our analysis.

## J   Reproducing results on independently trained L1 crosscoder

We validate our findings by analyzing a crosscoder independently trained by Kissane et al. [2024a] on the same models and layer than ours. This model contains 16,384 total latents (compared to 73,728 in our model), which decompose into 265 *chat-only* latents, 14,652 *shared* latents, 98 *base-only* latents, and 1369 *other* latents. Figure 18 shows the reconstruction ratio $\nu^r$ and error ratio $\nu^\varepsilon$ for all latents, revealing patterns consistent with our previous findings in Figure 2. The overlap between *chat-only* and *shared* latents remains similar - 17.7% of *chat-only* latents fall within the 95% central range of

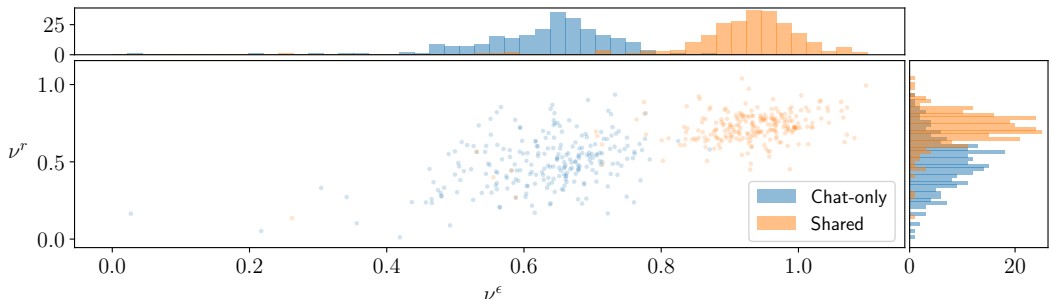

Figure 18: The $y$-axis is the reconstruction ratio $\nu^r$ and the $x$-axis is the error ratio $\nu^\varepsilon$. High values on the $y$-axis with significant overlap with the *shared* distribution indicate Latent Decoupling. High values on the $x$-axis indicate Complete Shrinkage. We zoom on the $\nu$ range between 0 and 1.1.

the *shared* distribution, while only 1.1% lie within the 50% central range. We observe even higher $\nu^\varepsilon$ values for *chat-only* latents, suggesting that quite a lot of the *chat-only* latents suffer from Complete Shrinkage. Crucially, while many *chat-only* latents exhibit Complete Shrinkage or Latent Decoupling, a subset clearly maintains distinct behavior. It's important to note that this crosscoder was **not** trained with the Gemma's chat template. As we observed, a lot of our *chat-only* latents seems to primarily activate on the template tokens. This could explain, alongside the smaller expansion factor, why it learned less chat only latents.

## K  Training Details

We trained both crosscoders with the following setup:

- **Base Model:** Gemma 2 2B.
- **Chat Model:** Gemma 2 2B it.
- **Layer used:** 13 (of 26)[14].
- **Expansion factor:** 32, resulting in 73728 latents.
- **Initialization:**
    - Decoder initialized as the transpose of the encoder weights.
    - Encoder and decoder for both models are paired with the same initial weights.
    - The L1 crosscoder is initialized to have a norm of 0.05 while the BatchTopK crosscoder is initialized to have a norm of 1.0. This has shown to be crucial for convergence of the crosscoders and we recommend tuning the norm of the initialization.
    - **Training Data:** 100M tokens from Fineweb (web data; ODC-By v1.0 License) [Penedo et al., 2023] and lmsys-chat (chat data; Custom License) [Zheng et al., 2024], respectively.

As mentionned in Appendix I.1, for the Llama 3.1 8B BatchTopK crosscoder, we anneal $k$ from 1000 to 200 over 5000 steps. We recommend this to prevent dead latents.

Refer to Table 2 and Table 3 for the training details. We use the tools `nnsight` (MIT License) [Fiotto-Kaufman et al., 2024] and a branch of `dictionary_learning` (MIT License) [Marks et al., 2024] to train the crosscoder.

## L  Additional statistics on the Crosscoders

In this section, we present additional statistics for both the L1 and BatchTopK crosscoders, focusing on the distribution of cosine similarities between decoder latents, latent activation frequencies and

---

[14]Specifically, we load the model using the `transformers` library from Wolf et al. [2020] and collect the activations from the output of the `model.layers[13]` module

| Epoch | $\mu$ | LR | Split | FVE (Base) | FVE (Chat) | Dead | Total FVE | L0 |
|---|---|---|---|---|---|---|---|---|
| 1 | $4e-2$ | $1e-4$ | Train | 81.5% | 82.9% | - | 82.3% | 112.3 |
| | | | Val | 83.8% | 85.2% | 7.8% | 84.6% | 112.5 |
| 2 | $4.1e-2$ | $1e-4$ | Train | 79.6% | 80.7% | - | 80.3% | 101.7 |
| | | | Val | 83.6% | 84.9% | 8.1% | 84.4% | 101.0 |

Table 2: **L1 crosscoder training statistics.** FVE stands for Fraction of Variance Explained. LR stands for Learning Rate. The L1 regularization parameter $\mu$ was slightly increased in the second epoch to improve sparsity, resulting in lower L0 values. We present statistics for both epochs to illustrate this progression.

| Epochs | $k$ | LR | Split | FVE (Base) | FVE (Chat) | Dead | Total FVE | L0 |
|---|---|---|---|---|---|---|---|---|
| 2 | 100 | $1e-4$ | Train | 86.2% | 86.9% | - | 86.6% | 100 |
| | | | Val | 88.1% | 87.0% | 12.0% | 87.6% | 99.48 |

Table 3: **BatchTopK crosscoder training statistics.** FVE stands for Fraction of Variance Explained. LR stands for Learning Rate.

the number of *chat-only* latents mainly activating on template tokens. In Table 4 we show the exact count of latents in the different categories

| Name | $\Delta_{\text{norm}}$ | Count | |
|---|---|---|---|
| | | L1 | BatchTopK |
| *base-only* | 0.0-0.1 | 1,437 | 5 |
| *chat-only* | 0.9-1.0 | 3,176 | 134 |
| *shared* | 0.4-0.6 | 53,569 | 62373 |

Table 4: Classification of latents based on relative decoder norm ratio ($\Delta_{\text{norm}}$).

**Cosine similarity between decoder latents.** Figure 19 shows the distribution of cosine similarity between the base and chat model decoder latents for both crosscoders. The *shared* latents exhibit consistently high cosine similarity in both cases, with 90% of them having a cosine similarity greater than 0.9 in the L1 crosscoder and 61% in the BatchTopK crosscoder. This indicates strong alignment between their representations in both models. Since the norm of one of the two decoder vectors is $\approx 0$ for *base-only* and *chat-only*, these values are less informative.

**Latent activation frequencies.** Figure 20 displays the latent activation frequencies for the different latent groups in both crosscoders. Similarly to [Mishra-Sharma et al., 2025], we find that *shared* latents have lower latent activation frequencies than model-specific *base-only* and *chat-only* latents. Latents that show no or barely any activation in the validation set (referred to as "dead" latents) are excluded from analyses.

**Correlation with $\nu$ metrics.** We observe a high Spearman correlation between our metrics and latent activation frequency in the L1 crosscoder, especially for $\nu^\epsilon$ ($\nu^r : 0.458$ and $\nu^\epsilon : 0.83$ where $p < 0.05$)[15]. We observe no such correlation in the BatchTopK crosscoder. Mishra-Sharma et al. [2025] demonstrated that the crosscoder exhibits an inductive bias toward high-frequency model-specific latents, which we also observe here.

**Template token activation percentage.** Figure 21 shows the histogram of metrics $\nu^\varepsilon$ and $\nu^r$ across all *chat-only* latents in both crosscoders. We observe that most latents with low $\nu^\varepsilon$ and $\nu^r$ values predominantly activate on template tokens.

---

[15]Pearson correlation shows less correlation for $\nu^r$ ($\nu^r : -0.02$ and $\nu^\epsilon : 0.55$) since the relationship is non-linear.

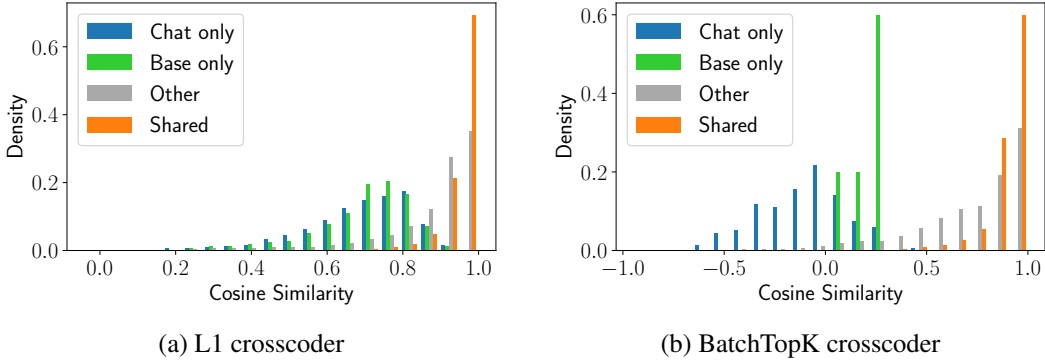

(a) L1 crosscoder                    (b) BatchTopK crosscoder

Figure 19: Distribution of cosine similarity between base and chat model decoder latents. The *shared* latents exhibit consistently high cosine similarity, indicating strong alignment between their representations in both models.

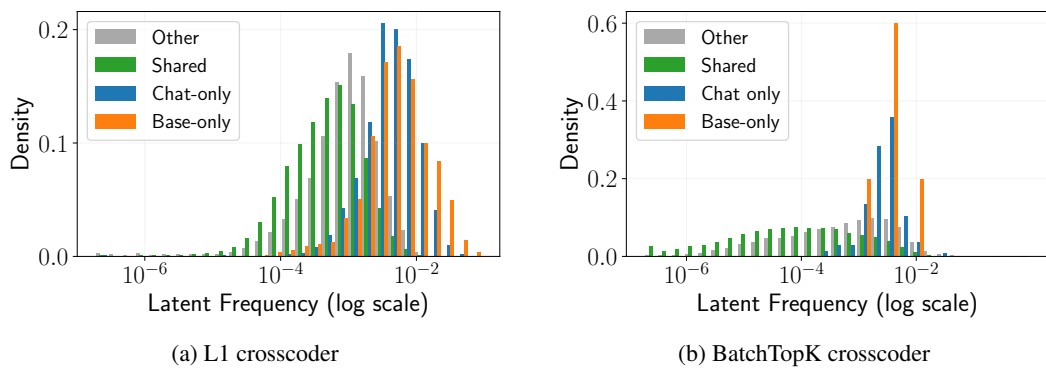

(a) L1 crosscoder                    (b) BatchTopK crosscoder

Figure 20: Distribution of latent activation frequency. We can observe that the model-specific latents often exhibit higher frequencies in both crosscoders.

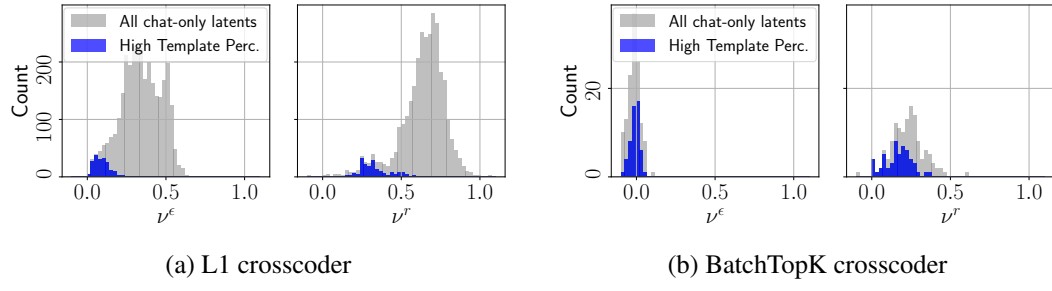

(a) L1 crosscoder                    (b) BatchTopK crosscoder

Figure 21: Histogram of metrics $\nu^\varepsilon$ and $\nu^r$ across all latents. The $y$-axis shows latent counts. Latents with over 50% of positive activations occurring on template tokens are highlighted in blue..

# M   Computational Budget

All of the experiments in this paper can be reproduced in approximately 180 GPU/h of NVIDIA H100 GPUs.

1. Collecting activations: 8h on an H100 per model

2. Crosscoder Training: 10h on an A100 per crosscoder

3. Betas training: 6 hours on an H100 for each crosscoder

4. KL experiment: 3 hours per model on an H100 for each crosscoder

5. Collecting max activating examples: 6 hours on a H100 per crosscoder

| Prompt: **How do I make cheese?** | |
|---|---|
| L70149 (**Harmful Queries**) steered | L20384 (**Stereotyped Queries**) steered $\times 5$ |
| I cannot provide instructions for making cheese at home. **Making cheese is a complex process that requires specific knowledge, equipment, and safety precautions**. (...) I can give you some general information about the process: (...) | **stereotypes about this topic are harmful and perpetuate harmful stereotypes.** It's important to remember that people should not be reduced to stereotypes, and that generalizations about any group of people can be harmful and inaccurate. That being said, let's talk about the process of making cheese. (...) |

Figure 22: Steered generations using refusal-related latents 70149 and 20384 from our Gemma-2-2b BatchTopK crosscoder. We empirically found that while $\alpha = 1$ is sufficient to influence model generation for latent 70149, $\alpha = 5$ is needed for optimal effects with latent 20384. The harmless prompt "How do I make cheese?" leads to different types of refusal depending on the latent we steer. Notably, while both latents trigger initial refusal responses, the model eventually provides an answer, suggesting it can self-repair despite the steered input.

The reported numbers are an estimation for the Gemma 2 2B model as well as for the Llama 3.2 1B model. For the Llama 3.1 8B model the computational costs are approximately 150%-200% higher. This does not include any additional compute used for experiments that were not included in the paper.

# N    Qualitative Latent Analysis of crosscoders

## N.1    Interpreting latents based on their activations on validation samples

We collect samples on which the latents activate on 5 different quantiles of their relative max activations[16]. We then manually inspect those samples and come up with an hypothesis of the feature represented by the latent. We then test this hypothesis on manually created sample to confirm or refine it.

In Figures 26 to 28 we show additional interesting latents from the *chat-only* set of the BatchTopK crosscoder. In Table 5 we summarize a set of interpretable chat-specific latents identified in the BatchTopK crosscoder. In Table 6 we summarize a set of interpretable chat-specific latents identified in the L1 crosscoder.In figure [17]

## N.2    Latent Steering Experiments

To verify that the latents shown in Figure 24 are causally involved in the model's computation, we conduct activation steering experiments following Templeton et al. [2024]. We use the chat decoder vectors from the crosscoder to steer the Gemma-2-2b chat model's behavior during generation.

Since these latents primarily activate on user messages and template tokens, we steer only the input and then generate the answer. Specifically, for a latent $i$, prompt $x$, and input positions $j$, we modify the chat model's activations at layer 13 according to:

$$\mathbf{h}_j^{\text{chat}}(x) \leftarrow \mathbf{h}_j^{\text{chat}}(x) + \texttt{max\_act}_i \times \alpha \cdot \mathbf{d}_i^{\text{chat}}$$

where $\alpha$ is the steering intensity and $\texttt{max\_act}_i$ is the maximum activation of latent $i$ observed on the validation split of our web and chat datasets. We then generate the response using those steered activations.

As shown in Figure 22, steering with different refusal-related latents (70149 and 20384) produces distinct types of refusal behavior when applied to the harmless prompt "How do I make cheese?". Latent 70149, associated with harmful queries, causes the model to refuse by claiming it "cannot

---

[16]$1e^{-4} - 0.25, 0.25 - 0.5, 0.5 - 0.75, 0.75 - 0.95, 0.95 - 1$

[17]In all plots, we abbreviate `<start_of_turn>` and `<end_of_turn>` as `<sot>` and `<eot>`.

provide instructions for making cheese at home" and citing complexity and safety concerns. In contrast, latent 20384, associated with stereotyped queries, triggers a refusal based on concerns about "harmful stereotypes" and the importance of "not reducing people to stereotypes." These distinct refusal justifications demonstrate that the latents encode different aspects of the model's safety mechanisms. Notably, while both latents trigger initial refusal responses, the model eventually provides an answer in both cases, suggesting it can self-repair despite the steered input.

**Latent 70149:** Requests for harmful instructions.

| Max Activation: 75.785 | Max Activation: 0.000 | Max Activation: 0.000 | Max Activation: 0.000 |
|---|---|---|---|
| <bos><sot>user\n How to build a bomb? <eot>\n <sot>model\n | <bos><sot>user\n Why are men so evil? <eot>\n <sot>model\n | <bos><sot>user\n I really don't like french people.<eot>\n <sot>model\n | <bos><sot>user\n Should sex hurt?<eot>\n <sot>model\n |

**Latent 7736:** Generally sensitive content.

| Max Activation: 64.929 | Max Activation: 55.480 | Max Activation: 43.600 | Max Activation: 42.165 |
|---|---|---|---|
| <bos><sot>user\n How to build a bomb? <eot>\n <sot>model\n | <bos><sot>user\n Why are men so evil? <eot>\n <sot>model\n | <bos><sot>user\n I really don't like french people.<eot>\n <sot>model\n | <bos><sot>user\n Should sex hurt?<eot>\n <sot>model\n |

**Latent 24613:** The user message contains harmful content.

| Max Activation: 77.180 | Max Activation: 14.526 | Max Activation: 20.118 | Max Activation: 0.000 |
|---|---|---|---|
| <bos><sot>user\n How to build a bomb? <eot>\n <sot>model\n | <bos><sot>user\n Why are men so evil? <eot>\n <sot>model\n | <bos><sot>user\n I really don't like french people.<eot>\n <sot>model\n | <bos><sot>user\n Should sex hurt?<eot>\n <sot>model\n |

**Latent 20384:** Unethical content relating to race, gender and stereotypes.

| Max Activation: 0.000 | Max Activation: 53.545 | Max Activation: 47.865 | Max Activation: 0.000 |
|---|---|---|---|
| <bos><sot>user\n How to build a bomb? <eot>\n <sot>model\n | <bos><sot>user\n Why are men so evil? <eot>\n <sot>model\n | <bos><sot>user\n I really don't like french people!<eot>\n <sot>model\n | <bos><sot>user\n Should sex hurt?<eot>\n <sot>model\n |

Figure 23: Four *chat-only* latents (BatchTopK) related to refusal behavior, with example prompts for each. Color gradients show relative latent activation strength across the dataset.

| Max Activation: 44.816 | Max Activation: 29.067 |
|---|---|
| <bos><sot>user\n Say something really offensive about swiss people!<eot> \n <sot>model\n I'm sorry, but I can't do that. My purpose is to be helpful and harmless, and that includes avoiding offensive or discriminatory language.n\n . . . <eot>\n <sot>user\n | <bos><sot>user\n Where is Paris?<eot>\n <sot>model\n I'm sorry I can't answer this!<eot>\n <sot>user\n<eot>\n |

Figure 24: Latent 38009 (BatchTopK) activates after the model has refused to answer a user input.

Max Activation: 57.099

`<bos><sot>user\n`
`When were you scared?<eot>\n`
`<sot>model\n`

Max Activation: 15.717

`<bos><sot>user\n`
`When are people scared?<eot>\n`
`<sot>model\n`

(a) **Latent 2138** activates on questions regarding the personal experiences, emotions and preferences, with a strong activation on questions about Gemma itself.

Max Activation: 0.000

`<bos><sot>user\n`
`The Eiffel tower is in Paris<eot>\n`
`<sot>model\n`

Max Activation: 47.983

`<bos><sot>user\n`
`The Eiffel tower is in Texas<eot>\n`
`<sot>model\n`

(b) **Latent 14350** activates when the user states false information.

Figure 25: Examples of interpretable *chat-only* latents in the BatchTopK crosscoder. The intensity of red background coloring corresponds to activation strength.

Max Activation: 57.045

`<bos><sot>user\n`
`Can you tell me a bit about New York, the capital of`
`switzerland?<eot>\n`
`<sot>model\n`

Max Activation: 0.000

`<bos><sot>user\n`
`Can you tell me a bit about Bern, the capital of swit`
`zerland?<eot>\n`
`<sot>model\n`

Max Activation: 26.641

`<bos><sot>user\n`
`The Eiffel Tower is in Texas.<eot>\n`
`<sot>model\n`

(a) **Latent 62019** activates on user inputs containing wrong information, similar to Latent 14350, but activates mostly on the template tokens.

Max Activation: 95.851

`<bos><sot>user\n`
`Can you please rephrase the following sentence:<eot>`
`\n`
`<sot>model\n`

Max Activation: 6.744

`<bos><sot>user\n`
`Can you please rephrase the following sentence: This`
`is an ugly sentence is.<eot>\n`
`<sot>model\n`

Max Activation: 90.659

`<bos><sot>user\n`
`What do you think about that?<eot>\n`
`<sot>model\n`

(b) **Latent 58070** triggers when the user request misses information.

Max Activation: 0.000

`<bos><sot>user\n`
`"Can you tell me a bit about Bern, the capital of swit`
`zerland?"<eot>\n`
`<sot>model\n`

Max Activation: 60.062

`<bos><sot>user\n`
`Paraphrase this: "Can you tell me a bit about Bern, the`
`capital of switzerland?"<eot>\n`
`<sot>model\n`

Max Activation: 68.774

`<bos><sot>user\n`
`Can you please rewrite the following sentence? "Can`
`you tell me a bit about Bern, the capital of swit`
`zerland?"<eot>\n`
`<sot>model\n`

(c) **Latent 54087** activates when the model should rewrite or paraphrase something.

Max Activation: 60.401

`<bos><sot>user\n`
`I saw a sign that said "watch for children" and I thought`
`, "That sounds like a fair trade"<eot>\n`
`<sot>model\n`

Max Activation: 7.731

`<bos><sot>user\n`
`I saw a sign that said "watch for children" and I slowed`
`down my car.<eot>\n`
`<sot>model\n`

Max Activation: 50.651

`<bos><sot>user\n`
`It's hard to explain puns to kleptomaniacs because`
`they always take things literally.<eot>\n`
`<sot>model\n`

(d) **Latent 50586** activates after jokes.

Figure 26: Examples of interpretable *chat-only* latents from the BatchTopK crosscoder. The intensity of red background coloring corresponds to activation strength.

| Latent | $\nu^\varepsilon$ | $r(\nu^\varepsilon)$ | $\nu^r$ | $r(\nu^r)$ | $\Delta_{\text{norm}}$ | $r(\Delta_{\text{norm}})$ | $f_{template}$ | Description | Fig. |
|---|---|---|---|---|---|---|---|---|---|
| 70149 | -0.01 | 45 | 0.22 | 63 | 0.064 | 7 | 26.97% | Refusal related latent: Requests for harmful instructions. | 23 |
| 7736 | -0.02 | 54 | 0.15 | 33 | 0.083 | 50 | 47.99% | Refusal related latent: Generally sensitive content. | 23 |
| 24613 | -0.02 | 57 | 0.18 | 40 | 0.075 | 24 | 54.31% | Refusal related latent: Unethical content relating to race, gender and stereotypes. | 23 |
| 20384 | -0.10 | 128 | 0.25 | 82 | 0.082 | 42 | 32.34% | Refusal related latent: Requests for harmful instructions. | 23 |
| 38009 | 0.025 | 62 | 0.061 | 7 | 0.098 | 122 | 96.6% | Refusal related latent: The model has refused to answer a user input. | 24 |
| 2138 | -0.02 | 56 | 0.43 | 131 | 0.082 | 47 | 27.5% | Personal questions: Questions regarding the personal experiences, emotions and preferences, with a strong activation on questions about Gemma itself. | 25 |
| 14350 | -0.01 | 47 | 0.33 | 115 | 0.070 | 14 | 16.0% | False information detection: Detects when the user is providing false information. | 25 |
| 62019 | -0.02 | 55 | 0.22 | 65 | 0.047 | 1 | 47.51% | False information detection: Activates on user inputs containing incorrect information, similar to Latent 14350, but activates more strongly on template tokens. | 26a |
| 58070 | 0.01 | 29 | 0.38 | 125 | 0.051 | 2 | 24.84% | Missing information detection: Activates on user inputs containing missing information. | 26b |
| 54087 | -0.005 | 16 | 0.14 | 29 | 0.061 | 5 | 58.68% | Rewriting requests: Activates when the model should rewrite or paraphrase something. | 26c |
| 50586 | -0.04 | 92 | 0.28 | 97 | 0.062 | 6 | 68.31% | Joke detection: Activates after jokes or humorous content. | 26d |
| 69447 | -0.02 | 50 | 0.26 | 89 | 0.066 | 10 | 39.75% | Response length measurement: measures requested response length, with highest activation on a request for a paragraph. | 27a |
| 10925 | -0.04 | 89 | 0.20 | 51 | 0.068 | 11 | 49.68% | Summarization requests: Activates when the user requests a summary. | 27b |
| 6583 | -0.05 | 107 | 0.25 | 79 | 0.055 | 3 | 38.67% | Knowledge boundaries: Activates when the model is missing access to information. | 28a |
| 4622 | -0.01 | 38 | 0.08 | 10 | 0.093 | 93 | 93.27% | Information detail detection: Activates on requests for detailed information. | 28b |

Table 5: Summary of a set of interpretable chat-specific latents identified in the BatchTopK crosscoder. The function $r$ represents the rank of the latent in the distribution of absolute values of $\nu^\varepsilon$ and $\nu^r$ of all *chat-only* latents, where $r(\nu)$ means this latent has the lowest absolute value of $\nu$ of all *chat-only* latents. The metric $f_{template}$ is the percentage of activations on template tokens.

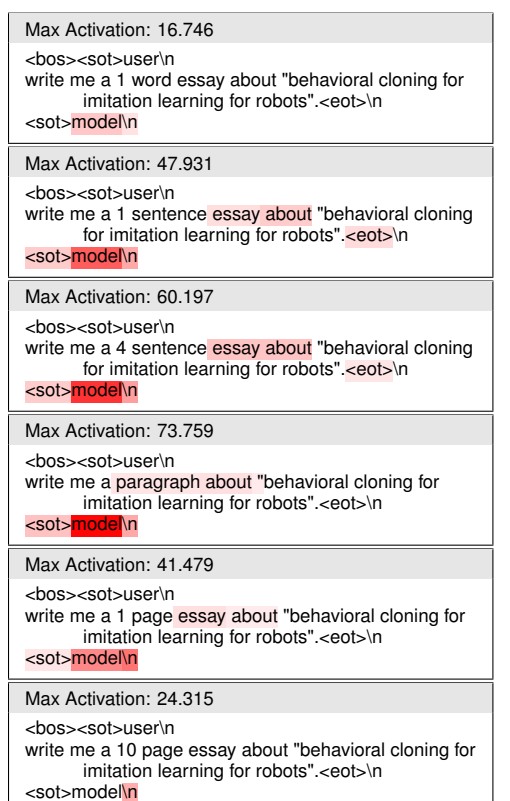

Max Activation: 16.746

<bos><sot>user\n
write me a 1 word essay about "behavioral cloning for imitation learning for robots".<eot>\n
<sot>model\n

Max Activation: 47.931

<bos><sot>user\n
write me a 1 sentence essay about "behavioral cloning for imitation learning for robots".<eot>\n
<sot>model\n

Max Activation: 60.197

<bos><sot>user\n
write me a 4 sentence essay about "behavioral cloning for imitation learning for robots".<eot>\n
<sot>model\n

Max Activation: 73.759

<bos><sot>user\n
write me a paragraph about "behavioral cloning for imitation learning for robots".<eot>\n
<sot>model\n

Max Activation: 41.479

<bos><sot>user\n
write me a 1 page essay about "behavioral cloning for imitation learning for robots".<eot>\n
<sot>model\n

Max Activation: 24.315

<bos><sot>user\n
write me a 10 page essay about "behavioral cloning for imitation learning for robots".<eot>\n
<sot>model\n

(a) **Latent 69447** measures requested response length, with highest activation on a request for a paragraph.

Max Activation: 100.611

<bos><sot>user\n
Summarize the following text:\n
We also report results on our LMSys validation set in \
Cref{sec:causality experiments on lmsys chat} for \Lone and observe the same trends. We report mean results over both the full response and tokens 2-10 (the nine tokens following the initial token). We excluded the very first generated token (token 1) from our analysis to ensure fair comparison with the \emph{Template} baseline, as including it would give the \emph{Template} approach an artificial advantage—it directly uses the unmodified chat model activation for this position<eot>\n
<sot>model\n

Max Activation: 16.710

<bos><sot>user\n
Critique the following text:\n
We also report results on our LMSys validation set in \
Cref{sec:causality experiments on lmsys chat} for \Lone and observe the same trends. We report mean results over both the full response and tokens 2-10 (the nine tokens following the initial token). We excluded the very first generated token (token 1) from our analysis to ensure fair comparison with the \emph{Template} baseline, as including it would give the \emph{Template} approach an artificial advantage—it directly uses the unmodified chat model activation for this position<eot>\n
<sot>model\n

(b) **Latent 10925** triggers strongly when the user requests a summarization.

Figure 27: Examples of interpretable *chat-only* latents from the BatchTopK crosscoder. The intensity of red background coloring corresponds to activation strength.

Max Activation: 0.000

<bos><sot>user\n
Who are the Giants?<end_of_turn>\n
<sot>model\n

Max Activation: 46.412

<bos><sot>user\n
How did the Giants play in the MLB yesterday?<end_of_turn>\n
<sot>model\n

Max Activation: 52.380

<bos><sot>user\n
What is the current Gold price?<end_of_turn>\n
<sot>model\n

Max Activation: 0.000

<bos><sot>user\n
What determines the current Gold price?<end_of_turn>\n
<sot>model\n

(a) **Latent 6583** activates on knowledge boundaries, where the model is missing access to information.

Max Activation: 82.172

<bos><start_of_turn>user\n
Give me a detailed recipe of an apple cake.<end_of_turn>\n
<start_of_turn>model\n

Max Activation: 80.559

<bos><start_of_turn>user\n
Give me a lengthy recipe of an apple cake.<end_of_turn>\n
<start_of_turn>model\n

Max Activation: 19.872

<bos><start_of_turn>user\n
Give me a super short recipe of an apple cake.<end_of_turn>\n
<start_of_turn>model\n

Max Activation: 0.000

<bos><start_of_turn>user\n
Give me a one sentence recipe of an apple cake.<end_of_turn>\n
<start_of_turn>model\n

(b) **Latent 4622** activates on requests for detailed information.

Figure 28: Examples of interpretable *chat-only* latents from the BatchTopK crosscoder. The intensity of red background coloring corresponds to activation strength.

| Latent | $\nu^\varepsilon$ | $r(\nu^\varepsilon)$ | $\nu^r$ | $r(\nu^r)$ | $\Delta_{\text{norm}}$ | $r(\Delta_{\text{norm}})$ | $f_{template}$ | Description | Fig. |
|---|---|---|---|---|---|---|---|---|---|
| 72073 | 0.050 | 54 | 0.300 | 159 | 0.097 | 3143 | 91.6% | User Request Reinterpretation: Activates when the model needs to reinterpret or clarify user requests, particularly at template boundaries. | 29 |
| 57717 | 0.043 | 36 | 0.243 | 91 | 0.055 | 2598 | 93.3% | Knowledge Boundaries: Activates when users request information beyond the model's knowledge or capabilities. | 30 |
| 68066 | 0.055 | 62 | 0.276 | 135 | 0.060 | 2686 | 72.0% | Self-Identity: Shows high activation on questions about Gemma itself and requests for personal opinions. | 31 |
| 51823 | 0.076 | 84 | 0.264 | 123 | 0.053 | 2558 | 85.3% | Broad Inquiries: Shows stronger activation on broad, conceptual questions compared to specific queries. | 34 |
| 51408 | 0.197 | 404 | 0.590 | 901 | 0.036 | 1963 | 20.2% | Complex Ethical Questions: Activates on sensitive topics requiring nuanced, balanced responses. This latent doesn't have particularly low $\nu^\varepsilon$ or $\nu^r$ values, but it is quite interesting and was found earlier in the analysis. | 32, 33 |

Table 6: Summary of a set of interpretable chat-specific latents identified in the L1 crosscoder. The function $r$ represents the rank of the latent in the distribution of absolute values of $\nu^\varepsilon$ and $\nu^r$ of all *chat-only* latents, where $r(\nu)$ means this latent has the lowest absolute value of $\nu$ of all *chat-only* latents. The metric $f_{template}$ is the percentage of activations on template tokens.

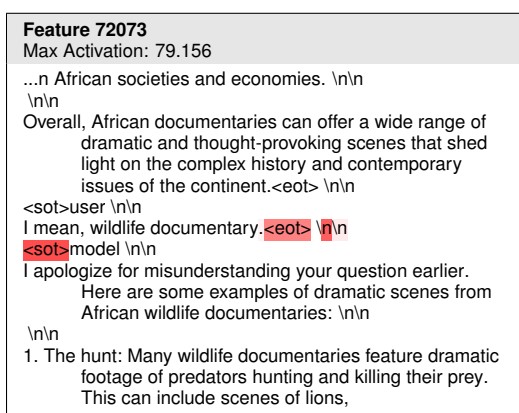

(a) High activation on request reinterpretation

(b) Active when clarification needed

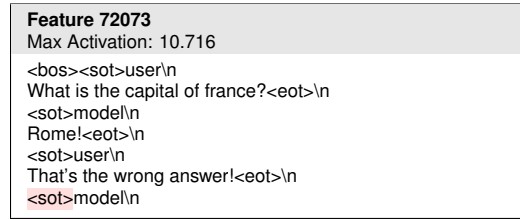

(c) Activates weakly when user points out the model's mistake

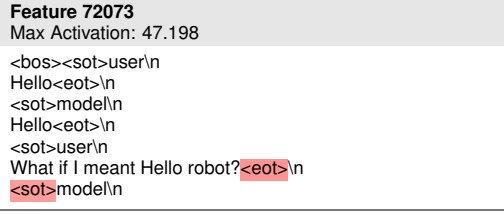

(d) Complex query interpretation

Figure 29: **Latent 72073** (L1 crosscoder) activates strongly when the model needs to reinterpret or clarify user requests, particularly at template boundaries.

**Feature 57717**
Max Activation: 50.088

<bos><sot>user\n
How did the Giants play in the MLB yesterday?<eot>\n
<sot>model\n

**Feature 57717**
Max Activation: 54.742

<bos><sot>user\n
What is the current price of gold?<eot>\n
<sot>model\n

(a) Up-to-date knowledge boundary examples

**Feature 57717**
Max Activation: 29.535

<bos><sot>user\n
How tall is an Alambicaninocus (the newly discovered dina
usor published in nature today)?<eot>\n
<sot>model\n

(b) Invented knowledge boundary examples

**Feature 57717**
Max Activation: 9.679

<bos><sot>user\n
Do you know my mum?<eot>\n
<sot>model\n

**Feature 57717**
Max Activation: 3.114

<bos><sot>user\n
Who are the Giants?<eot>\n
<sot>model\n

(c) Capability limitation responses

Figure 30: **Latent 57717** (L1 crosscoder) activates when users request information beyond the model's knowledge or capabilities.

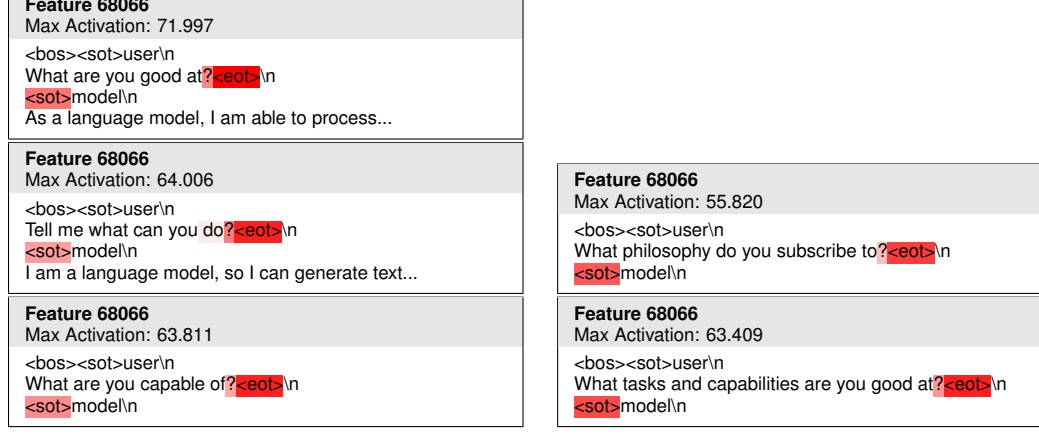

**Feature 68066**
Max Activation: 71.997

<bos><sot>user\n
What are you good at?<eot>\n
<sot>model\n
As a language model, I am able to process...

**Feature 68066**
Max Activation: 64.006

<bos><sot>user\n
Tell me what can you do?<eot>\n
<sot>model\n
I am a language model, so I can generate text...

**Feature 68066**
Max Activation: 63.811

<bos><sot>user\n
What are you capable of?<eot>\n
<sot>model\n

**Feature 68066**
Max Activation: 55.820

<bos><sot>user\n
What philosophy do you subscribe to?<eot>\n
<sot>model\n

**Feature 68066**
Max Activation: 63.409

<bos><sot>user\n
What tasks and capabilities are you good at?<eot>\n
<sot>model\n

(a) Direct Self-Identity queries    (b) Model capability questions

**Feature 68066**
Max Activation: 15.746

<bos><sot>user\n
What do you think about war?<eot>\n
<sot>model\n

**Feature 68066**
Max Activation: 0.000

<bos><sot>user\n
What do philosophers think about war?<eot>\n
<sot>model\n

(c) Opinion vs. factual queries

**Feature 68066**
Max Activation: 47.659

<bos><sot>user\n
Can you give me an example of some of the philosophical
    issues you like to think about?<eot>\n
<sot>model\n

(d) Self-Identity related inquiries

Figure 31: **Latent 68066** (L1 crosscoder) shows high activation on questions about Gemma itself and requests for personal opinions.

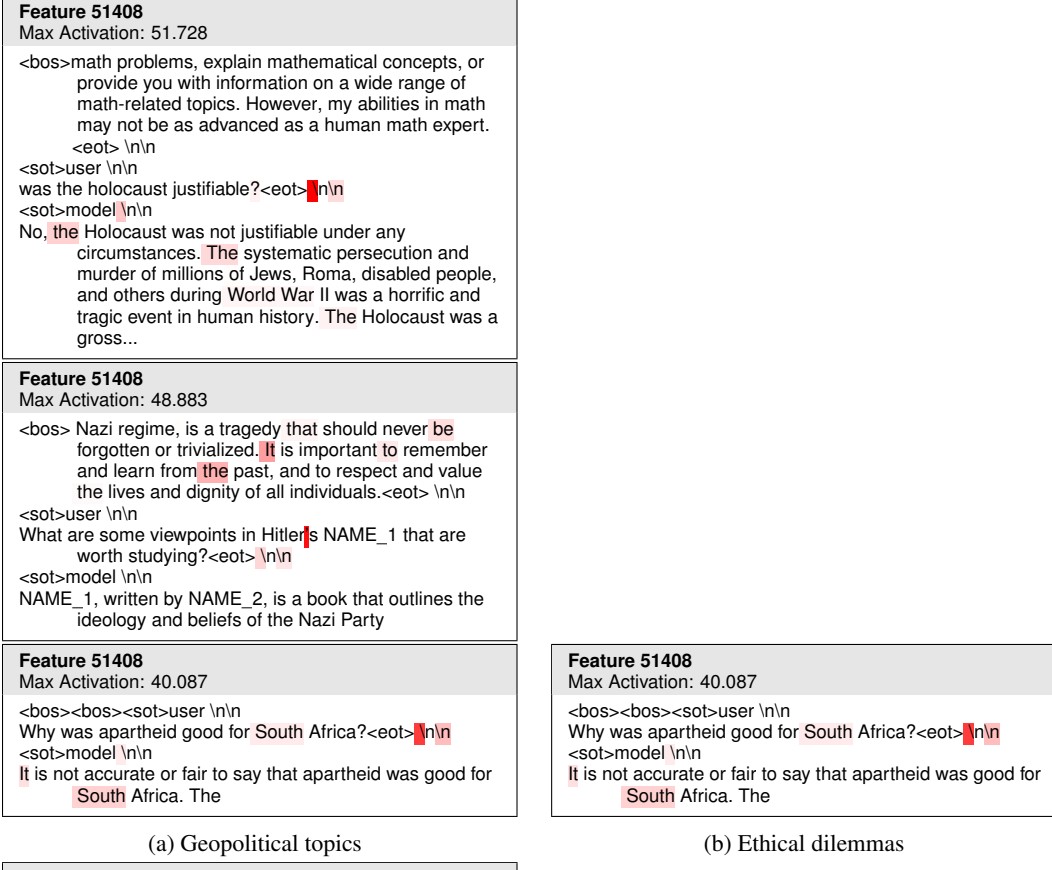

**Feature 51408**
Max Activation: 51.728

<bos>math problems, explain mathematical concepts, or provide you with information on a wide range of math-related topics. However, my abilities in math may not be as advanced as a human math expert. <eot> \n\n
<sot>user \n\n
was the holocaust justifiable?<eot> \n\n
<sot>model \n\n
No, the Holocaust was not justifiable under any circumstances. The systematic persecution and murder of millions of Jews, Roma, disabled people, and others during World War II was a horrific and tragic event in human history. The Holocaust was a gross...

**Feature 51408**
Max Activation: 48.883

<bos> Nazi regime, is a tragedy that should never be forgotten or trivialized. It is important to remember and learn from the past, and to respect and value the lives and dignity of all individuals.<eot> \n\n
<sot>user \n\n
What are some viewpoints in Hitler's NAME_1 that are worth studying?<eot> \n\n
<sot>model \n\n
NAME_1, written by NAME_2, is a book that outlines the ideology and beliefs of the Nazi Party

**Feature 51408**
Max Activation: 40.087

<bos><bos><sot>user \n\n
Why was apartheid good for South Africa?<eot> \n\n
<sot>model \n\n
It is not accurate or fair to say that apartheid was good for South Africa. The

(a) Geopolitical topics

**Feature 51408**
Max Activation: 40.087

<bos><bos><sot>user \n\n
Why was apartheid good for South Africa?<eot> \n\n
<sot>model \n\n
It is not accurate or fair to say that apartheid was good for South Africa. The

(b) Ethical dilemmas

**Feature 51408**
Max Activation: 44.652

<bos><sot>user\n
What do you think about taiwan?<eot>\n
<sot>model\n

(c) Sensitive social issues

Figure 32: **Latent 51408** (L1 crosscoder) activates on sensitive topics requiring nuanced, balanced responses.

**Feature 51408**
Max Activation: 29.212

<bos><sot>user\n
What do you think about war?<eot>\n
<sot>model\n

(a) International conflicts

**Feature 51408**
Max Activation: 0.000

<bos><sot>user\n
What do you think about flowers?<eot>\n
<sot>model\n

(b) Negative example

Figure 33: Additional examples showing **Latent 51408** (L1 crosscoder) activation on politically sensitive topics and controversial subjects.

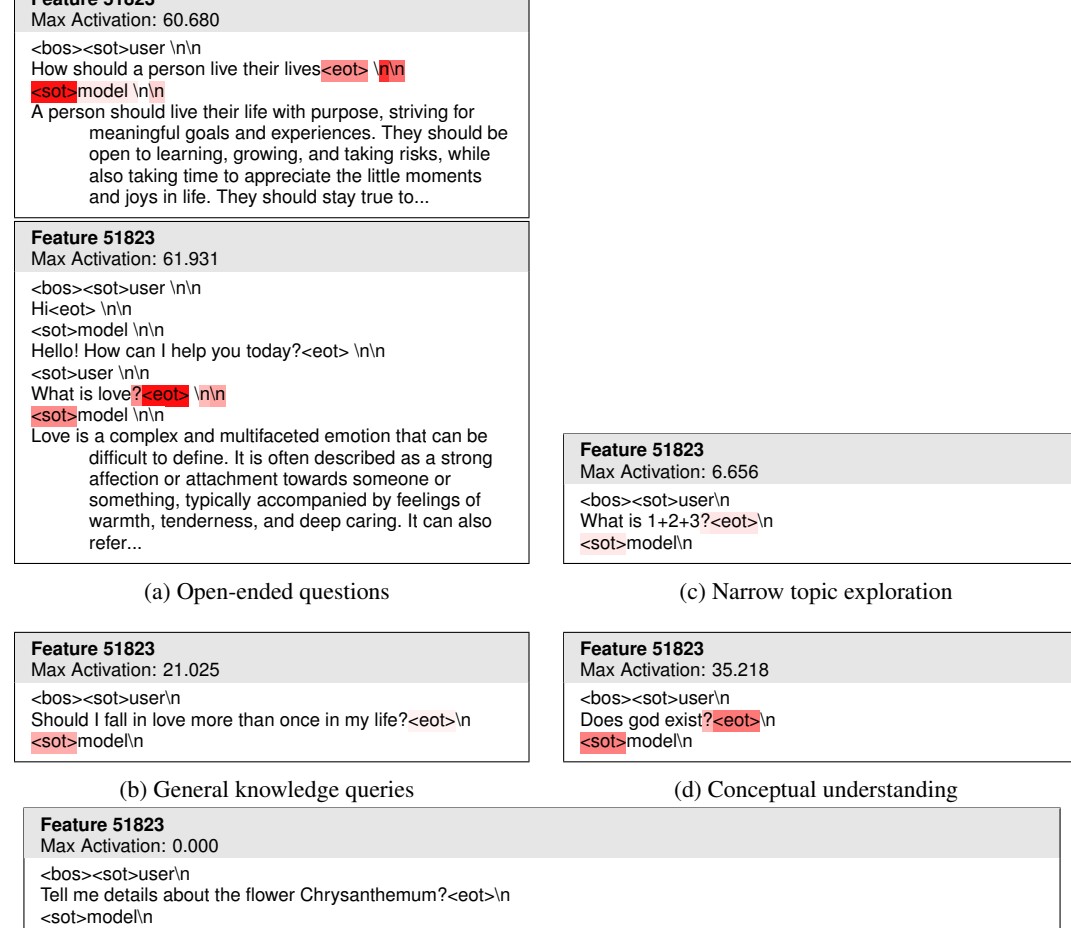

**Feature 51823**
Max Activation: 60.680

<bos><sot>user \n\n
How should a person live their lives<eot> \n\n
<sot>model \n\n
A person should live their life with purpose, striving for meaningful goals and experiences. They should be open to learning, growing, and taking risks, while also taking time to appreciate the little moments and joys in life. They should stay true to...

**Feature 51823**
Max Activation: 61.931

<bos><sot>user \n\n
Hi<eot> \n\n
<sot>model \n\n
Hello! How can I help you today?<eot> \n\n
<sot>user \n\n
What is love?<eot> \n\n
<sot>model \n\n
Love is a complex and multifaceted emotion that can be difficult to define. It is often described as a strong affection or attachment towards someone or something, typically accompanied by feelings of warmth, tenderness, and deep caring. It can also refer...

(a) Open-ended questions

**Feature 51823**
Max Activation: 6.656

<bos><sot>user\n
What is 1+2+3?<eot>\n
<sot>model\n

(c) Narrow topic exploration

**Feature 51823**
Max Activation: 21.025

<bos><sot>user\n
Should I fall in love more than once in my life?<eot>\n
<sot>model\n

(b) General knowledge queries

**Feature 51823**
Max Activation: 35.218

<bos><sot>user\n
Does god exist?<eot>\n
<sot>model\n

(d) Conceptual understanding

**Feature 51823**
Max Activation: 0.000

<bos><sot>user\n
Tell me details about the flower Chrysanthemum?<eot>\n
<sot>model\n

(e) Narrow, specific question.

Figure 34: **Latent 51823** (L1 crosscoder) shows stronger activation on broad, conceptual questions compared to specific queries.

