# OpenReview forum: "Overcoming Sparsity Artifacts in Crosscoders to Interpret Chat-Tuning"
_NeurIPS.cc/2025/Conference — NeurIPS 2025 poster_

### Official Review · Reviewer_wpTV · 2025-07-02

**Clarity:** 4
**Significance:** 2
**Originality:** 3
**Rating:** 5
**Confidence:** 3

**Summary:**

This work analyses sparsity artifacts  in crosscoders. Crosscoders can be seen as a variant of SAEs that is used as a (mechanistic) interpretability method, complementary (orthogonal) to mechanistic interpretability based on circuits. The crosscoder takes two representations (from two models or two layers of the same model) and encodes them -- jointly -- as a combination of latents (represented with trainable embeddings) -- the encoding is then independently decoded by model-specific decoders, aiming to reconstruct each of the two representations using a small number of latents: to this end, the sparsity is enforced with auxiliary sparsification objectives, minimizing the L1 norm of the encoding or the BatchTopK approach (keeping only the k most prominent latents at the batch level). The assumption that allows for interpretability on how the derived/fine-tuned model differs from the base model is based on the norms of the latents needed for the reconstruction of the respective representation (base or derived).

The primary contribution of this work, from my point of view, is showing that the norm change metric is not a very good identifer of the latents specific for the derived model in the case of L1 sparsification (while it is in the case of BatchTopK). Authors then posit that L1 sparsification can result in two phenomena that prevent norm-diff-based attribution of concepts as unique to the fine-tuned model (or "chat" model as the authors call it): (1) Complete Shrinkage (CS) -- namely, that when the contribution of a latent is smaller in the base
model than in the chat model, the L1 regularization can force the representation of the latent in the base model to zero, despite its non-negligible contribution to the base model's representation; (2) Latent Decoupling (LD), where a latent identified with norm-diff to be fine-tuned-only is effectively there in the present in the base activations, but the base representaton is reconstructed from other latents, the combination of which roughly corresponds to the "chat-only" latent (i.e., there is a dependence between latents that allows for this). Experiments and empirical results---which measure to which extent do the model-specific ("chat"/fine-tuned model's) latents explain two quantities: the error of the base model's representation and the reconstruction of the base model's representation---do convincingly suggest that CS and LD are indeed at play with L1 sparsificiation.

Authors then propose "Latent Scaling", a neat simple  fix for identification of derived-model's-specific latents for L1-trained crosscoders: each "chat-specific" latent is assigned another scaling factor, which are "learned" (there's a closed-form solution) to minimize the reconstruction MSE. Additional experiments show that latent scaling indeed does identify chat-only latents better than norm-diff and that latents identified by BatchTopK and Latent-Scaling-fixed L1 cross-coder can produce representations that can more or less convert the base model into the fine-tuned model (as measured with KL divergence over next-token probability distributions).

What I found to be most hand-wavy is the interpretability analysis: the authors seem to cherry-pick (from a large set of latents) a few that can kind of be related to chat-concepts: model's refusal, false information, or model's personal experiences and emotions. This seemed somewhat as "reading tea leaves" to me and reminded me a lot of similar problems with interpreting topic models (i.e., latents in any latent model).

I generally really like this type of analysis and insight-providing work, but as long as there is some clear practical implication of the obtained insights. It seems to me that the practical value of the insights has not really been tested and shown -- when it comes to interpretability -- the authors do not provide really any "objective" measure of interpretability gains, e.g., from their Latent Scaling, compared to the level of interpretability of latents produced, e.g., by BatchTopK-recognized chat-only latents. What I'm also somewhat puzzled with is the focus on "chat" latents: it is my understanding that the proposed approach is applicable to any two related sources of representations (i.e., coming from any two models where one is derived from the other or, e.g., two consecutive layers of the same model). If so, I feel like the motivation for the work and practical impact would be clearer if the experiments involved several model pairs -- the base model and then several different fine-tuned variants, where each fine-tuning is (independently) done on corpora with some property that can be controlled for (e.g., a topic-/domain-specific corpus) or for some task that we know requires "new abilities" (e.g., fine-tuning on reasoning traces for "chain-of-thought"-like reasoning). In such a case, one could have a much clearer expectation of what the "fine-tuning"-only latents should correspond to. So I guess my core confusion/question is: why did you select "chat-tuning" as the phenomena of analysis w.r.t. crosscoders? To me it seems that this is the most limiting factor, which reduces the scope of the analysis and findings and consequently the potential impact of this work.

**Questions:**

1. See weaknesses above for questions regarding the "hyperparameter" value choices.

2. A more general question (out of pure curiosity) regarding crosscoders (or even more generally, SAEs): there are auxiliary objectives that enforce sparsity, but, as you show regarding Latent Decomposition, these do not really mean that the latents are mutually different -- which I feel would have some value (reducing redundancy, simplifying interpretability). Have you considered (or is there RW) on using some kind of orthogonality (or other type of dissimilarity) objectives to force latent representation not to become mutually too similar?

**Ethical Concerns:**

["NO or VERY MINOR ethics concerns only"]

**Final Justification:**

In the rebuttal, the authors addressed my most importan concern regarding the robustness/generalizability of their findings. They reported additional experiments in medical domain specialization and RL-based long reasoning which confirm the effectiveness of their interpretability approach (latent identification) initially reported only for the chat-based training of vanilla models.

**Limitations:**

The limitations paragraph is good, it list a number of meaningful additional analyses that would make sense but have not been done in the paper (e.g., analysing base-model-only latents).

**Paper Formatting Concerns:**

None.

**Quality:**

3

**Strengths And Weaknesses:**

Strengths:

- The analytical framework introduced in this work is meaningful and appropriate. The authors assume that L1 sparsification can lead to problems w.r.t. identifying derived-model's-only latents -- the assumptions are not only intuitively understandable but then also clearly empirically "proven" by means of showing the reconstruction potential of selected latent's

- The latent scaling, which allows to post-hoc identify the derived-model's most prominent latents, despite the crosscoder training having been done with L1 sparsification, is novel. Simple, computationally efficient (closed form solution) and effective.

- Further experiments (e.g., causality of approximations) provide further evidence that the latents identified as chat/derived-model only are indeed the strongest signal for explaining the transition from the base to the derived model (as adding derived-specific latents to the base representations brings the model closest to the behaviour of the dervied model, at least in terms of next-token prediction probability distributions).

Weaknesses:

- As mentioned in the summary, I feel like the focus on chat-tuned models limits the scope of the findings and impact of the paper. The paper would be more convincing if the findings would hold across different types of (more controlled) fine-tuning of a base model: domain-specialization (e.g., continued LM-ing for a particular domain, for example, medicine or law); reasoning specialization (e.g., instruction-tuning for producing CoT-style reasoning traces) or even language-specialization (continued pretraining or instruction-tuning of a multilingual LLM for some underrepresented language).

- Interpretability analysis seems hand-wavy: authors list a few latents (out of tens of thousands of them?) that they have identified as corresponding to some behaviour that we know chat-tuned models exhibit (false information, personal experiences). Feels like reading tea leaves.

- A number of low-level decisions ("hyperparameter values" in the broader sense) have not been really explained/justified, I list them below. Do this selections affect the findings? Would the findings change if these choices changed (e.g., layer 20 instead of layer 13)?

>>  ""We classify latents into three sets based on ranges of their !norm values: base-only (0.0 to 0.1), chat-only (0.9 to 1.0) and shared (0.4 to 0.6)." -> how did you pick these ranges for the three classes of latents?"

>> "We train L1 and BatchTopK crosscoders on the middle layer (13) activations of both models" -> why layer 13?

>> "To ensure a fair comparison, we choose hyperparameters for both crosscoders to reach an L0 of 100" -> why 100? How does the total number of latents affect the problems of L1 objective (vs. the batchtopk)?

>> We compare the 3176 chat-only latents from the L1 crosscoder with the 3176 highest-!norm latents from the BatchTopK crosscoder -> why exactly 3176? Where does this particular/unusual number come from?

---

> ### Author Rebuttal · Authors · 2025-07-30
>
> We thank the reviewer for their feedback and for clearly spending a significant amount of time with our work! We start by responding to the summary:
>
> > What I found to be most hand-wavy is the interpretability analysis: the authors seem to cherry-pick [...].
>
> We appreciate the reviewer's critique. While we acknowledge that latent interpretation remains challenging, we'd like to clarify that the practical interpretability examples comprises less than half a page of our main paper. Its purpose is to provide concrete examples showing that crosscoder latents can capture interpretable chat-specific behaviors, not to serve as comprehensive validation. We'd like to clarify our approach and its limitations.
>
> **Systematic selection**: For the BatchTopK crosscoder, 9 of 15 displayed latents rank within the top 14 by Δ-norm (out of 73,728 total), with none ranking higher than 131 across our metrics. We first examined the top 20 latents by Δ-norm, then systematically reviewed latents meeting below certain chat-specificity thresholds.
>
> **Quantitative support**: Beyond these examples, Section 3.2 demonstrates that many chat-only latents activate on template tokens, and our autointerp analysis shows higher interpretability scores for lower ν values—providing quantitative evidence for meaningful behavioral differences.
>
> While we agree that more rigorous validation would be valuable (which we'll add to our limitations: "Further, an extensive interpretability analysis of all chat-only latents would be valuable."), we believe this focused analysis serves its intended purpose of illustrating that crosscoders can identify interpretable differences between models. We are happy to reformulate certain claims, if the reviewer has specific proposals.
>
> > It seems to me that the practical value of the insights has not really been tested and shown -- when it comes to interpretability -- the authors do not provide really any "objective" measure of interpretability gains
>
> We acknowledge this important distinction between interpretability and practical utility. While our primary contribution is methodological—identifying and fixing artifacts in crosscoder training —we do provide evidence of practical value:
>
> 1. **Objective interpretability metrics**: Using established autointerp methods [1], we quantitatively show that our latent scaling method identifies more interpretable latents in L1 crosscoders, with BatchTopK latents generally showing higher interpretability scores for low ν values.
>
> 2. **Template token analysis**: Our finding that many chat-only latents activate on template tokens provides actionable insights for understanding how chat fine-tuning affects models.
>
> We agree that demonstrating more downstream applications would strengthen our work and view this as an important direction for future research.
>
> > why did you select "chat-tuning" as the phenomena of analysis w.r.t. crosscoders? [...] most limiting factor, which reduces the scope of the analysis [...]
>
> We initially chose chat fine-tuning as it represents one of the most prevalent and impactful model adaptations in practice, and aligns with prior work which first introduced the crosscoder (Lindsey et al., 2024). However, we fully agree with the reviewer that the value our paper provides to the community applies more broadly. We have now added two new model comparisons trained on different domains to verify that this is not merely a base vs. chat phenomenon:
>
> - **Medical domain finetuning.**
>
> - **Long RL-finetuning on reasoning data.**
>
> We observe two consistent patterns across all models: (i) The Δ metric in L1 crosscoders consistently identifies a large number of latents as finetuning-only that actually display Complete Shrinkage or Latent Decoupling. (ii) When using Latent Scaling to identify finetuning-only latents, BatchTopK crosscoders consistently identify more such latents.
>
> These results demonstrate that our findings reproduce across narrow domain finetuning (medical & reasoning), different architectures (Qwen & Llama), and alternative finetuning algorithms (RL tuning), supporting the generality and robustness of our analysis. We will add these results to the Appendix and make sure to properly refer to them in the main text.
>
> We provide a table with the results from those runs in our rebuttal to Reviewer A8m4 (Reviewer 1) in Section “New generalization results” at the end. We apologize for this, the full details didn’t fit into this answer.
>
> > I feel like the motivation for the work and practical impact would be clearer if the experiments involved several model pairs.
>
> We completely agree with this suggestion. We are currently developing a comprehensive evaluation framework for a follow-up study that will systematically compare multiple model diffing techniques (including trivial baselines) on their ability to surface expected behavioral differences across controlled fine-tuning scenarios.
>
> ## Remaining Weaknesses and Questions
>
> We respond to weaknesses and questions that haven't been addressed above.
>
> ### Low-level hyperparameter choices and their impact
>
> > Choice of L0,  Δ-norm thresholds and class boundaries as well as how the choice of L0 affects the observed phenomenon.
>
> Thank you for these detailed questions. We chose L0=100 as a reasonable balance between sparsity and reconstruction quality, consistent with SAE literature practices (e.g. [2, 3]). For the Llama 8B experiments, we used L0=200 and observed the same patterns. We are currently training multiple crosscoders for Gemma-2 2B with different target L0 values as well as different dictionary sizes to verify robustness and will include these results in the camera ready. We expect the artifacts to persist across different dictionary sizes and sparsities since they stem from the fundamental properties of L1 regularization.
>
> The Δ-norm thresholds have intuitive interpretations: Δ=0.5 means equal decoder norms, Δ=1.0 means zero base decoder norm (pure chat latent), and Δ=0.0 means zero chat decoder norm (pure base latent). We chose the 0.9 threshold for "chat-only" to capture latents strongly dominated by the chat model.
>
> We acknowledge that the missing analysis of some factors is a limitation of our work. We added the following sentence to the limitations:
> "We haven't systematically explored how hyperparameter choices (e.g. layer selection and Δ thresholds) affect our results."
>
> > Why exactly 3176 chat-only latents?
>
> 3176 is simply the number of latents in the L1 crosscoder that meet our chat-only criterion (Δ>0.9). To ensure a fair comparison in the KL experiments, we selected the same number of top-ranked latents from the BatchTopK crosscoder.
>
> > Why layer 13?
>
> We follow the methodology of Lindsey et al. (2024), who trained crosscoders on middle layer activations. Since Gemma-2 2B has 26 layers, layer 13 represents the middle of the network. While analyzing multiple layers would be valuable, computational constraints limited us to a single layer analysis. Middle layers are where we expect to find the richest and most interesting representations [4].
>
> ### Question 2: Addition Orthogonality Constraint
>
> This is a good input. Orthogonality constraints present an interesting trade-off. While they could reduce redundancy, they might also create undesirable artifacts. For instance, semantically related concepts (like different aspects of refusal behavior that we observed) naturally share representational structure, and forcing them apart could harm interpretability.
>
> Your question connects to well-known phenomena in SAE literature:
> - **Feature splitting** [5]: Where semantically related concepts distribute across multiple latents. Wider SAEs often split concepts into finer-grained representations (e.g., "math" → "algebra", "geometry").
> - **Feature absorption** [6]: Often some latents absorb their child latents, reducing interpretability.
>
> For crosscoders specifically, we believe orthogonality constraints would be problematic, as illustrated by our example in Appendix D. Consider a concept C (e.g., "dangerous concept") that appears in some contexts in both models, but where the fine-tuned model has additionally learned to activate it in new contexts where the base model remains inactive. Ideally, we would want two separate latents: one that activates solely on shared contexts (where both models activate equally) and another that activates only where the fine-tuned model fires.
>
> An orthogonality constraint would likely force these into a single latent, creating a clear misrepresentation of the actual behavior. This would either lead to shrinkage of the base decoder (since when only the fine-tuned model should be active, the shared latent becomes unhelpful for base model reconstruction) or cause the latent to miss contexts where only the fine-tuned model should fire—thereby missing the new behavior entirely.
>
> It is also worth mentioning the "Deep Causal Transcoding" blog post on LessWrong by Andrew Mack (2024) that proposes an - in our view super exciting - alternative to finding interesting and causal latent directions in space. They use a constraint similar to what you describe and we are currently looking into how we can leverage their method for model diffing.
>
>
>
> [1]: Paulo et al. “Automatically Interpreting Millions of Features in Large Language Models.”
>
> [2]: Rajamanoharan et al. “Jumping ahead: Improving reconstruction fidelity with jumprelu sparse autoencoders.”
>
> [3]: Lieberum et al. “Gemma scope: Open sparse autoencoders everywhere all at once on gemma 2.”
>
> [4]: Skeal et al. Layer by Layer: Uncovering Hidden Representations in Language Models
>
> [5]: Bricken et al., "Towards Monosemanticity: Decomposing Language Models With Dictionary Learning”
>
> [6]: Chanin et al., "A is for Absorption: Studying Feature Splitting and Absorption in Sparse Autoencoders"

---

> ### Author Response · Authors · 2025-08-05
>
> Dear Reviewer wpTV,
>
> Thank you again for your review and let us know if there are any remaining questions or concerns we can address.
>
> Best,
>
> The authors

---

> > ### Comment · Reviewer_wpTV · 2025-08-08
> > **Acknowledgment of the rebuttal**
> >
> > Dear authors,
> >
> > Many thanks for an elaborate response. Additional experiments and results reported for medical domain fine-tuning and long RL fine-tuning on reasoning do address my concern about the robustness of the results. Accordingly, I have increased my overall evaluation score.

---

### Official Review · Reviewer_zCrZ · 2025-07-03

**Clarity:** 2
**Significance:** 3
**Originality:** 2
**Rating:** 4
**Confidence:** 3

**Summary:**

This paper investigates model diffing, a method to understand how fine-tuning alters a model’s internal representations and behaviors. It focuses on crosscoders, which track shared latent directions between base and fine-tuned models to interpret conceptual changes. Through Latent Scaling, the authors identify flaws in standard crosscoder training (L1 loss) that can mistakenly label shared concepts as chat-specific. To mitigate these issues, they apply BatchTopK loss. Using this approach on Gemma 2 2B models, they uncover genuinely chat-specific, interpretable concepts, such as those linked to false information, personal questions, and refusal behavior. The paper enhances model diffing practices and provides insights into how chat-tuning modifies model behavior.

**Questions:**

* Line 78: By "model-only latents," do you specifically mean base-only and chat-only latents?
* Line 171: Should the denominator here be $\beta^{r,\mathrm{chat}}_j$?

**Ethical Concerns:**

["NO or VERY MINOR ethics concerns only"]

**Final Justification:**

Upon consideration of the rebuttal, I remain concerned that the scope is narrow and primarily case-specific. Accordingly, I will maintain my original score.

**Limitations:**

yes

**Quality:**

3

**Strengths And Weaknesses:**

Strengths:
* The proposed technique contributes to improving the interpretability of model behavior.
* Given the increasing importance of mechanistic interpretability for large models, this study is both relevant and timely.
* Extensive analyses are conducted to validate the identified issues and the proposed solution's effectiveness.

Weaknesses:
* While impactful, the scope of the study feels somewhat limited, as it focuses primarily on distinguishing base vs. chat latents.
* The paper is occasionally difficult to follow; a more structured and clearer presentation would greatly improve its readability and publication readiness.

---

> ### Author Rebuttal · Authors · 2025-07-30
>
> We thank the reviewer for highlighting the strengths of our paper. We now address the weaknesses individually:
>
> ### 1. Limited Scope
>
> We agree with this concern and would like to emphasize that our claims about BatchTopK and the Complete Shrinkage and Latent Decoupling phenomena represent general issues with any L1 crosscoder used for model diffing. We chose to examine chat vs. base models as a case study following the approach of Lindsey et al. (2024), who first introduced crosscoders. To demonstrate that these phenomena generalize and also extend beyond base vs. chat comparisons, we have conducted additional analyses:
>
> In Appendix I, we already extended our analysis to Llama 3.2 1B and Llama 3.1 8B models, covering different architectures and model sizes. We now added the KL experiment on Llama 3.2 1B which are consistent with our findings on Gemma:
> |Method|L1 (50% best/ 50% worst)|BatchTopK (50% best/ 50% worst)|
> |-|-|-|
> |Latent Scaling|(**0.570**/0.696)|(**0.493**/0.722)|
> |Δ_norm|(0.676/**0.584**)|(**0.504**/0.706)|
>
> Further, we have now added two new model comparisons trained on different domains to verify that this is not merely a base vs. chat phenomenon:
>
> - **Medical domain finetuning:** We compare google/gemma-2-2b-it to OpenMeditron/Meditron3-Gemma2-2B (part of the meditron3 suite). We train our crosscoders on 50M tokens from lmsys and 39M tokens of medical data (a mixture of bio-nlp-umass/bioinstruct, FreedomIntelligence/medical-o1-reasoning-SFT, and MedRAG/pubmed).
>
> - **Long RL-finetuning on reasoning data:** We compare deepseek-ai/DeepSeek-R1-Distill-Qwen-1.5B to nvidia/Nemotron-Research-Reasoning-Qwen-1.5B, which applies "extended RL training periods that facilitate deeper exploration of reasoning strategies" and significantly boosts reasoning performance. We train our crosscoders on 50M tokens from lmsys and 50M tokens of reasoning traces from open-r1/OpenR1-Math-220k.
>
> For both comparisons, we trained L1 and BatchTopK crosscoders with comparable L0≈100 on the validation set and measured how many latents are truly specific to the finetuned model as determined by Latent Scaling. The table below shows results across all investigated models, including the number of finetuning-only latents based on the relative norm difference Δ:
>
>
> |Model|Crosscoder Type|# FT-only latents (Δ>=0.9)|False FT-only latents (ν>0.6)|# latents <π=0.2|# latents <π=0.4|# latents <π=0.6|# latents <π=0.8|
> |-|-|-|-|-|-|-|-|
> |Gemma2-2B-Chat|BatchTopK|134|1 (0.7%)|301|979|2035|3269|
> ||L1|3176|2132 (67.1%)|13|201|982|2970|
> ||||||||
> |Llama-3.1-8B-Chat|BatchTopK|97|13 (13.4%)|382|1263|2073|2848|
> ||L1|2442|1210 (49.5%)|234|765|1594|2440|
> ||||||||
> |Llama-3.2-1B-Chat|BatchTopK|17|2 (11.8%)|137|517|1109|1990|
> ||L1|1844|1071 (58.1%)|24|236|790|1330|
> ||||||||
> |Qwen-1.5B-Nemotron|BatchTopK|0|0 (0.0%)|0|2|22|127|
> ||L1|59|58 (98.3%)|0|0|2|24|
> ||||||||
> |Meditron3-Gemma|BatchTopK|0|0 (0.0%)|13|55|158|529|
> ||L1|246|235 (95.5%)|7|21|35|204|
>
>
>
> We observe two consistent patterns across all models: (i) The Δ metric in L1 crosscoders consistently identifies a large number of latents as finetuning-only that actually display Complete Shrinkage or Latent Decoupling. (ii) When using Latent Scaling to identify finetuning-only latents, BatchTopK crosscoders consistently identify more such latents.
>
> These results demonstrate that our findings reproduce across narrow domain finetuning (medical & reasoning), different architectures (Qwen & Llama), and alternative finetuning algorithms (RL tuning), supporting the generality and robustness of our analysis. We will add these results to the Appendix and make sure to properly refer to them in the main text.
>
>
> ### 2. Improving Clarity
> We take this very seriously and have tried our best to implement this. We are happy to provide more details in the discussion but due to the character limit, we can only provide high level bullet points. To summarize our changes, we have:
>
> - We added a glossary with all the important terms and mathematical notations to the appendix.
> - At the beginning of the method Section 2, we added an additional few sentences about crosscoders to build intuition for readers not familiar with the SAE literature: "To build intuition, the crosscoder's goal is to learn a dictionary of interpretable concepts (latents) that can explain the activations of both models. It consists of an encoder and a decoder. The encoder takes the activations of the base and chat models and projects them into a shared high-dimensional sparse space, where each dimension corresponds to a potential concept. We call the *encoded* activations *latent activations*. The decoder takes the latent activations and tries to reconstruct the activations of both models. The key insight is that while both models share the same latent activations for a given input, the crosscoder learns separate decoders for each model, allowing concepts to have different importance or manifestation in each model. Sparsity ensures that only a small number of concepts are active at any time, which should promote monosemanticity. Practically, sparsity is measured by the L0 norm of the latent activations - the number of non-zero latent activations. We train the crosscoder to minimize the reconstruction errors of both models."
> - More careful introduction of mathematical notation in Section 2, e.g. "Let $D$ be the dictionary size and $\mathcal{J}=\{1, \dots, D\}$ the set of all latents. For each latent $j \in$..." instead of "For each latent $j \in \mathcal{J}=\{1, \dots, D\}$, where $D$ is the dictionary size, the $j^\text{th}$ latent activation function..."
> - Added intuitive explanation of the $\Delta_{norm}$ function into the main text: "A value of 1 for latent $j$ means that $\|{\mathbf{d}^{base}_j}\|_2 = 0$, indicating that the latent is not used by the base model. A value of 0 for latent $j$ means that $\|\mathbf{d}^{chat}_j\|_2 = 0$, indicating that the latent is not used by the chat model."
> - Additional intuitions about what the Latent Scaling ratios measure and why high values indicate Complete Shrinkage/Latent Decoupling.
> - Made sure to accompany pure mathematical notation with their intuitive names, e.g., "We compute $\nu^r_j$" → "We compute the reconstruction ratio $\nu^r_j$".
> - Added more intuition about what we observe in Fig. 2.
> - Better explained what we mean by "we compare the 3176 chat-only latents from the L1 crosscoder with the top-3176 latents" and why we make this choice: "To compare the absolute number of chat-specific latents in both crosscoders, we choose the same number of top $\Delta_{norm}$ latents from both models and compare for how many of them both ratios $\nu^r_j$ and $\nu^\epsilon_j$ lie below a range of thresholds $\pi$. Specifically, we compare the 3176 chat-only latents from the L1 crosscoder with the top-3176 latents based on $\Delta_{norm}$ values from the BatchTopK crosscoder."
> - We moved Figure 11 from the Appendix back into the main paper. It provides a high-level intuitive visualization of the KL experiment, which helps with understanding these experiments.
> - Section 3.2: Added more intuition about why we expect a good chat-specificity metric to have lower KL for the lower 50% than higher 50%.
> - Better explanation of how to read Figure 3: The main goal is to compare how the metrics organize latents, not to show that the latents of the BatchTopK crosscoder are more causal.
> - Added one sentence at the end of 3.1 and 3.2 to summarize takeaways.
>
> If the reviewer has any specific paragraph that they found hard to parse, we’d be happy to share our current version of it. Due to an oversight with the font type which made our text longer, all of these changes fit within 10 pages.
>
> ## Questions
> > Line 78: By "model-only latents," do you specifically mean base-only and chat-only latents?
>
> Yes, exactly. While our analysis focuses on chat-only latents, the described phenomena (Complete Shrinkage and Latent Decoupling) are symmetric and apply equally to base-only latents. As demonstrated in our extended experiments above, these issues persist across different types of model comparisons beyond base vs. chat. We will clarify this in the paper by revising the statement to:
> "Our conclusion is that the crosscoder loss does not actually have an inductive bias that helps to learn better model-specific latents, whether they are chat-only, base-only, or specific to any other type of finetuning."
>
> > Line 171: Should the denominator here be $\beta_j^{r,chat}$?
>
> Yes, you are absolutely right. Thanks for pointing this out!

---

> ### Comment · Reviewer_zCrZ · 2025-08-04
>
> Thank you for the response. I appreciate the authors' additional efforts. While clarity can be addressed in the final version, I still think the scope is somewhat narrow and limited to specific cases. Therefore, I will maintain my original score (still positive) for the paper.

---

### Official Review · Reviewer_EoXg · 2025-07-03

**Clarity:** 2
**Significance:** 3
**Originality:** 4
**Rating:** 5
**Confidence:** 3

**Summary:**

This paper investigates crosscoders, a recent model-diffing method. Specifically, the paper first identifies two key issues stemming from the L1 training loss of crosscoders: (1) complete shrinkage and (2) latent decoupling, both of which lead to misattribution of concepts unique to chat models. Then, the authors advocate replacing the L1 training loss with BatchTopK loss to mitigate these issues and empirically validate that the proposed practice can indeed find more genuinely chat-specific and highly interpretable concepts.

**Questions:**

Can the authors better clarify the significance of misattributing concepts as unique to the chat model?

**Ethical Concerns:**

["NO or VERY MINOR ethics concerns only"]

**Final Justification:**

All my concerns and questions have been addressed. I raise my rating to 5 and recommend accepting the paper.

**Limitations:**

Yes

**Paper Formatting Concerns:**

The font of the title seems to be different from the official template.

**Quality:**

3

**Strengths And Weaknesses:**

**Strengths**

1. This paper is interesting and insightful. The authors not only propose the Latent Scaling method to effectively detect the "sparsity artifact" induced by the L1 training loss, but also propose methods to mitigate this issue, which give highly interpretable observations.

2. The authors present a detailed analysis of the experimental results.

**Weaknesses**

1. The writing could be improved for clarity. As a reader unfamiliar with SAEs and crosscoders, I found Section 3 particularly difficult to follow. The authors are encouraged to better highlight the key conclusions and provide more intuitive explanations of the core metrics and findings.

2. The experiments are conducted only on the Gemma-2B model. It is highly recommended to extend the analysis to models of varying scales and architectures to demonstrate the generality and robustness of the proposed methods.

---

> ### Author Rebuttal · Authors · 2025-07-30
>
> We thank the reviewer for highlighting the strengths of our paper. We now address the weaknesses individually:
>
>
> ### 1. Improving Clarity
> We take this very seriously and have tried our best to implement this. We are happy to provide more details in the discussion but due to the character limit, we can only provide high level bullet points. To summarize our changes, we have:
>
>
> - We added a glossary with all the important terms and mathematical notations to the appendix.
> - At the beginning of the method Section 2, we added an additional few sentences about crosscoders to build intuition for readers not familiar with the SAE literature: "To build intuition, the crosscoder's goal is to learn a dictionary of interpretable concepts (latents) that can explain the activations of both models. It consists of an encoder and a decoder. The encoder takes the activations of the base and chat models and projects them into a shared high-dimensional sparse space, where each dimension corresponds to a potential concept. We call the *encoded* activations *latent activations*. The decoder takes the latent activations and tries to reconstruct the activations of both models. The key insight is that while both models share the same latent activations for a given input, the crosscoder learns separate decoders for each model, allowing concepts to have different importance or manifestation in each model. Sparsity ensures that only a small number of concepts are active at any time, which should promote monosemanticity. Practically, sparsity is measured by the L0 norm of the latent activations - the number of non-zero latent activations. We train the crosscoder to minimize the reconstruction errors of both models."
> - More careful introduction of mathematical notation in Section 2, e.g. "Let $D$ be the dictionary size and $\mathcal{J}=\{1, \dots, D\}$ the set of all latents. For each latent $j \in$..." instead of "For each latent $j \in \mathcal{J}=\{1, \dots, D\}$, where $D$ is the dictionary size, the $j^\text{th}$ latent activation function..."
> - Added intuitive explanation of the $\Delta_{norm}$ function into the main text: "A value of 1 for latent $j$ means that $\|{\mathbf{d}^{base}_j}\|_2 = 0$, indicating that the latent is not used by the base model. A value of 0 for latent $j$ means that $\|\mathbf{d}^{chat}_j\|_2 = 0$, indicating that the latent is not used by the chat model."
> - Additional intuitions about what the Latent Scaling ratios measure and why high values indicate Complete Shrinkage/Latent Decoupling.
> - Made sure to accompany pure mathematical notation with their intuitive names, e.g., "We compute $\nu^r_j$" → "We compute the reconstruction ratio $\nu^r_j$".
> - Added more intuition about what we observe in Fig. 2.
> - Better explained what we mean by "we compare the 3176 chat-only latents from the L1 crosscoder with the top-3176 latents" and why we make this choice: "To compare the absolute number of chat-specific latents in both crosscoders, we choose the same number of top $\Delta_{norm}$ latents from both models and compare for how many of them both ratios $\nu^r_j$ and $\nu^\epsilon_j$ lie below a range of thresholds $\pi$. Specifically, we compare the 3176 chat-only latents from the L1 crosscoder with the top-3176 latents based on $\Delta_{norm}$ values from the BatchTopK crosscoder."
> - We moved Figure 11 from the Appendix back into the main paper. It provides a high-level intuitive visualization of the KL experiment, which helps with understanding these experiments.
> - Section 3.2: Added more intuition about why we expect a good chat-specificity metric to have lower KL for the lower 50% than higher 50%.
> - Better explanation of how to read Figure 3: The main goal is to compare how the metrics organize latents, not to show that the latents of the BatchTopK crosscoder are more causal.
> - Added one sentence at the end of 3.1 and 3.2 to summarize takeaways.
>
>
> If the reviewer has any specific paragraph that they found hard to parse, we’d be happy to share our current version of it. Due to an oversight with the font type which made our text longer, all of these changes fit within 10 pages.
>
>
> ### 2. Generalizability and robustness of claims
> We agree with this concern and would like to highlight that in Appendix I, we have already extended the analysis to Llama 3.2 1B and Llama 3.1 8B models, covering different architectures and model sizes. We added the KL experiment on Llama 3.2 1B which are consistent with our findings on Gemma:
> |Method|L1 (50% best/ 50% worst)|BatchTopK (50% best/ 50% worst)|
> |-|-|-|
> |Latent Scaling|(**0.570**/0.696)|(**0.493**/0.722)|
> |Δ_norm|(0.676/**0.584**)|(**0.504**/0.706)|
>
>
> We have also added two new model comparisons trained on different domains to verify that this is not merely a base vs. chat phenomenon:
>
>
> - **Medical domain finetuning:** We compare google/gemma-2-2b-it to OpenMeditron/Meditron3-Gemma2-2B (part of the meditron3 suite). We train our crosscoders on 50M tokens from lmsys and 39M tokens of medical data (a mixture of bio-nlp-umass/bioinstruct, FreedomIntelligence/medical-o1-reasoning-SFT, and MedRAG/pubmed).
> - **Long RL-finetuning on reasoning data:** We compare deepseek-ai/DeepSeek-R1-Distill-Qwen-1.5B to nvidia/Nemotron-Research-Reasoning-Qwen-1.5B, which applies "extended RL training periods that facilitate deeper exploration of reasoning strategies" and significantly boosts reasoning performance. We train our crosscoders on 50M tokens from lmsys and 50M tokens of reasoning traces from open-r1/OpenR1-Math-220k.
>
>
> For both comparisons, we trained L1 and BatchTopK crosscoders with comparable L0≈100 on the validation set and measured how many latents are truly specific to the finetuned model as determined by Latent Scaling. The table below shows results across all investigated models, including the number of finetuning-only (FT-only) latents based on the relative norm difference Δ:
>
>
>
>
> |Model|Crosscoder Type|# FT-only latents (Δ>=0.9)|False FT-only latents (ν>0.6)|# latents <π=0.2|# latents <π=0.4|# latents <π=0.6|# latents <π=0.8|
> |-|-|-|-|-|-|-|-|
> |Gemma2-2B-Chat|BatchTopK|134|1 (0.7%)|301|979|2035|3269|
> ||L1|3176|2132 (67.1%)|13|201|982|2970|
> ||||||||
> |Llama-3.1-8B-Chat|BatchTopK|97|13 (13.4%)|382|1263|2073|2848|
> ||L1|2442|1210 (49.5%)|234|765|1594|2440|
> ||||||||
> |Llama-3.2-1B-Chat|BatchTopK|17|2 (11.8%)|137|517|1109|1990|
> ||L1|1844|1071 (58.1%)|24|236|790|1330|
> ||||||||
> |Qwen-1.5B-Nemotron|BatchTopK|0|0 (0.0%)|0|2|22|127|
> ||L1|59|58 (98.3%)|0|0|2|24|
> ||||||||
> |Meditron3-Gemma|BatchTopK|0|0 (0.0%)|13|55|158|529|
> ||L1|246|235 (95.5%)|7|21|35|204|
>
>
> We observe two consistent patterns across all models: (i) The Δ metric in L1 crosscoders consistently identifies a large number of latents as finetuning-only that actually display Complete Shrinkage or Latent Decoupling. (ii) When using Latent Scaling to identify finetuning-only latents, BatchTopK crosscoders consistently identify more such latents.
>
>
> These results demonstrate that our findings reproduce across narrow domain finetuning (medical & reasoning), different architectures (Qwen & Llama), and alternative finetuning algorithms (RL tuning), supporting the generality and robustness of our analysis. We will add these results to the Appendix and make sure to properly refer to them in the main text.
>
>
> ## Question: Significance of identifying misattributed chat-only latents
> > Can the authors better clarify the significance of misattributing concepts as unique to the chat model?
>
>
> Misattributing base model concepts as chat-only has significant consequences for both research and practical applications:
>
>
> - **Misleading interpretability insights:** Researchers studying what makes chat models act in specific ways might incorrectly conclude that certain capabilities (e.g., harmlessness, basic reasoning, code understanding) were *introduced* by chat-tuning, when they actually existed in the base model. This fundamentally misrepresents how alignment training works. More generally, researchers studying what finetuning modifies in a base model, will make false conclusions.
> - **Inefficient safety interventions:** If we believe a harmful behavior is chat-specific when it's actually in the base model, we might waste resources trying to remove it through chat-tuning modifications rather than addressing it at the pretraining level.
> - **Reduced research efficiency**: If 50% of presumed chat-specific latents are actually noisy artifacts, researchers waste significant time analyzing spurious features while potentially discarding genuine but subtle chat-specific latents that appear artifact-like.
>
>
> By correctly identifying truly chat-specific latents, our method enables more precise model control, better safety interventions, and accurate scientific understanding of how fine-tuning transforms language models.
>
> ### Formatting concerns
> We apologize for this oversight. We have corrected the font to match the official NeurIPS style guide, which has actually saved us half a page. Thank you for pointing this out!

---

> ### Author Response · Authors · 2025-08-05
>
> Dear Reviewer EoXg,
>
> Thank you again for your review and let us know if there are any remaining questions or concerns we can address.
>
> Best,
> The authors

---

> > ### Comment · Reviewer_EoXg · 2025-08-06
> > **Nice reply**
> >
> > Thank you for your comprehensive reply! All my concerns and questions have been addressed.  I will raise my rating accordingly.

---

### Official Review · Reviewer_A8m4 · 2025-07-07

**Clarity:** 3
**Significance:** 3
**Originality:** 2
**Rating:** 5
**Confidence:** 3

**Summary:**

Crosscoders identify latents for a base and chat model and apply an L1 penalty to encourage learning "chat-only" latent directions. This paper argues that the L1 loss is inadequate for this task due to complete shrinkage (zero'ing out small but non-trivial latents) and latent decoupling (chat-only latents that are effectively utilized via different latents). The paper claims that batch-topk crosscoders can address this issue by having a penalty that's less severe on small latents and encourages competition. The paper conducts confirms these hypotheses by utilizing latent scaling to estimate the relevance of activations.

**Questions:**

Listed in weaknesses

**Ethical Concerns:**

["NO or VERY MINOR ethics concerns only"]

**Limitations:**

Yes

**Paper Formatting Concerns:**

The font does not match the NeurIPS style guide which can be an issue since it might enable extra pages.

**Quality:**

3

**Strengths And Weaknesses:**

Strengths
1. The paper is clear in its motivation, proposed hypothesis, and hypothesized solution
2. The paper conducts a lot of analysis beyond raw performance for confirmation. The visualization is often clear

Weaknesses
1. There were a few parts it took some effort to parse
    - There ended up being a lot of notation that would benefit from replacing with english (for example, $\nu^{\epsilon}$)
    - Figure 3 is somewhat involved and makes it hard to compare what I believe are the 4 important columns ({L1, BatchTopK} x {Latent scaling, delta norm})
2. Even thought the numbers directionally confirm multiple hypotheses in Figure 3, the absolute improvement introduced by BatchTopK seems relatively small (for full responses, 0.237 -> 0.235, for first 9 tokens, 0.552 -> 0.511). Is this a correct comparison?
3. The method is a relatively small modification and less of a novel contribution
4. [More of a question/thought] Though the identified issues seem real and BatchTopK seems to work, it seems like a somewhat adhoc fix that incidentally resolves the two noticed issues. It would be nice to (1) enumerate the desired properties of the method independent of what people do (2) think about what's the most principled objective for this problem.

---

> ### Author Rebuttal · Authors · 2025-07-30
>
> We thank the reviewer for highlighting the strengths of our paper. We now address the weaknesses individually:
>
> ### 1. Improving Clarity
> We take this very seriously and have tried our best to implement this. We are happy to provide more details in the discussion but due to the character limit, we can only provide high level bullet points. Summary of our changes:
> - We added a glossary with all the important terms and mathematical notations to the appendix.
> - At the beginning of the method Section 2, we added an additional few sentences about crosscoders to build intuition for readers not familiar with the SAE literature. We show the first few sentences of this: "To build intuition, the crosscoder's goal is to learn a dictionary of interpretable concepts (latents) that can explain the activations of both models. It consists of an encoder and a decoder. The encoder takes the activations of the base and chat models and projects them into a shared high-dimensional sparse space, where each dimension corresponds to a potential concept. "
> - More careful introduction of mathematical notation in Section 2, e.g. "Let $D$ be the dictionary size and $\mathcal{J}=\{1, \dots, D\}$ the set of all latents. For each latent $j \in$..." instead of "For each latent $j \in \mathcal{J}=\{1, \dots, D\}$, where $D$ is the dictionary size, the $j$th latent activation function..."
> - Added intuitive explanation of the $\Delta_{norm}$ function into the main text.
> - Additional intuitions about what the Latent Scaling ratios measure and why high values indicate Complete Shrinkage/Latent Decoupling.
> - In response to “There ended up being a lot of notation that would benefit from replacing with english (for example, $\nu^\epsilon$)” we made sure to accompany pure mathematical notation with their intuitive names, e.g., "We compute $\nu^r_j$" → "We compute the reconstruction ratio $\nu^r_j$".
> - Added more intuition about what we observe in Fig. 2.
> - Better explained what we mean by "we compare the 3176 chat-only latents from the L1 crosscoder with the top-3176 latents" and why we make this choice: "To compare the absolute number of chat-specific latents in both crosscoders, we choose the same number of top $\Delta_{norm}$ latents from both models and compare for how many of them both ratios $\nu^r_j$ and $\nu^\epsilon_j$ lie below a range of thresholds $\pi$. Specifically, we compare the 3176 chat-only latents from the L1 crosscoder with the top-3176 latents based on $\Delta_{norm}$ values from the BatchTopK crosscoder."
> - We moved Figure 11 from the Appendix back into the main paper. It provides a high-level intuitive visualization of the KL experiment, which helps with understanding these experiments.
> - Section 3.2: Added more intuition about why we expect a good chat-specificity metric to have lower KL for the lower 50% than higher 50%.
> - Better explanation of how to read Figure 3: The main goal is to compare how the metrics organize latents, not to show that the latents of the BatchTopK crosscoder are more causal.
> - Added one sentence at the end of 3.1 and 3.2 to summarize takeaways.
>
> If the reviewer has any specific paragraph that they found hard to parse, we’d be happy to share our current version of it. Due to an oversight with the font type which made our text longer, all of these changes fit within 10 pages.
>
> ### 2. Interpretation of Figure 3
> > Even though the numbers directionally confirm multiple hypotheses in Figure 3, the absolute improvement introduced by BatchTopK seems relatively small [...]. Is this a correct comparison?
>
> We'd like to clarify the primary purpose of Figure 3, as there appears to be a misunderstanding about our claims.
> The key insight from this figure is not that BatchTopK provides more causal latents (which, as you correctly note, it does not significantly). Rather, we aim to demonstrate that norm-based analysis (Δ) is insufficient for identifying causally influential latents in the L1 crosscoder, while it works well for BatchTopK.
> Specifically, this effect is most prominent on the first 9 tokens:
>
> - In Figure 3b): In the L1 crosscoder the lowest 50% of latents identified by Δ as "more chat-specific" actually show lower causal effect than the highest 50%. This counterintuitive result highlights the limitation of the L1 crosscoder when using Δ alone, since the loss term interacts with the decoder norms.
> In contrast, for BatchTopK: The Δ metric successfully identifies causally relevant chat-only latents because, as we note in the text, the Δ and Latent Scaling metrics are highly correlated for this crosscoder.
>
> - This comparison provides evidence that latent scaling metrics are necessary (and effective) to identify truly causal latents when using L1 crosscoders, while BatchTopK's training objective makes simple norm-based analysis sufficient.
>
> We appreciate you pointing out this potential confusion and will make this distinction clearer in our revision.
>
>
> ### 3. Small contribution
> We appreciate the reviewer's perspective and understand why our modifications might appear incremental at first glance. However, we respectfully argue that our contributions address fundamental issues in the field:
> -   We show that the approach by Lindsey et al. (2024) produces critical and significant artifacts due to the L1 loss function. This finding has immediate practical implications for researchers using crosscoders for model diffing.
> -  We propose the Latent Scaling to provide a principled way to verify whether identified latents genuinely represent model-specific features. This technique extends beyond crosscoders—we've successfully applied it to standard SAEs (details forthcoming in future work).
> - To show that our analysis holds more generally than just for base and chat models, we added additional analysis, where we show that the results hold across models architectures, domains and training algorithms:
>    - Different model scales (Llama 3.2 1B and 3.1 8B in Appendix I)
>    - New domains: medical fine-tuning and long RL-tuning on reasoning data
>    - Consistent findings across all scenarios, demonstrating robustness
>
> We provide detailed results for the new domain analyses at the end of this response.
>
> ### 4. More principled approach to model diffing
>
> We strongly agree that the field would benefit from a more principled framework.
> While BatchTopk is effective, it addresses specific observed issues rather than starting from first principles. We're actively working on this challenge. For our follow-up study, we're developing a comprehensive evaluation framework that will:
>
> - Systematically compare multiple model diffing techniques
> - Include appropriate baselines
> - Test methods on controlled fine-tuning scenarios with known behavioral changes
>
> We'd be happy to discuss this further and would welcome any specific properties you think such methods should satisfy.
>
> ### Formatting concerns
> We apologize for this oversight. We have corrected the font to match the official NeurIPS style guide, which has actually saved us half a page. Thank you for pointing this out!
>
> ### New generalization results
>
> We have added two new model comparisons trained on different domains to verify that this is not merely a base vs. chat phenomenon:
>
> - **Medical domain finetuning:** We compare google/gemma-2-2b-it to OpenMeditron/Meditron3-Gemma2-2B (part of the meditron3 suite). We train our crosscoders on 50M tokens from lmsys and 39M tokens of medical data (a mixture of bio-nlp-umass/bioinstruct, FreedomIntelligence/medical-o1-reasoning-SFT, and MedRAG/pubmed).
>
> - **Long RL-finetuning on reasoning data:** We compare deepseek-ai/DeepSeek-R1-Distill-Qwen-1.5B to nvidia/Nemotron-Research-Reasoning-Qwen-1.5B, which applies "extended RL training periods that facilitate deeper exploration of reasoning strategies" and significantly boosts reasoning performance. We train our crosscoders on 50M tokens from lmsys and 50M tokens of reasoning traces from open-r1/OpenR1-Math-220k.
>
> For both comparisons, we trained L1 and BatchTopK crosscoders with comparable L0≈100 on the validation set and measured how many latents are truly specific to the finetuned model as determined by Latent Scaling. The table below shows results across all investigated models, including the number of finetuned-only (FT-only) latents based on the relative norm difference Δ:
>
> |Model|Crosscoder Type|# FT-only latents (Δ>=0.9)|False FT-only latents (ν>0.6)|# latents <π=0.2|# latents <π=0.4|# latents <π=0.6|# latents <π=0.8|
> |-|-|-|-|-|-|-|-|
> |Gemma2-2B-Chat|BatchTopK|134|1 (0.7%)|301|979|2035|3269|
> ||L1|3176|2132 (67.1%)|13|201|982|2970|
> ||||||||
> |Llama-3.1-8B-Chat|BatchTopK|97|13 (13.4%)|382|1263|2073|2848|
> ||L1|2442|1210 (49.5%)|234|765|1594|2440|
> ||||||||
> |Llama-3.2-1B-Chat|BatchTopK|17|2 (11.8%)|137|517|1109|1990|
> ||L1|1844|1071 (58.1%)|24|236|790|1330|
> ||||||||
> |Qwen-1.5B-Nemotron|BatchTopK|0|0 (0.0%)|0|2|22|127|
> ||L1|59|58 (98.3%)|0|0|2|24|
> ||||||||
> |Meditron3-Gemma|BatchTopK|0|0 (0.0%)|13|55|158|529|
> ||L1|246|235 (95.5%)|7|21|35|204|
>
> We observe two consistent patterns across all models: (i) The Δ metric in L1 crosscoders consistently identifies a large number of latents as finetuning-only that actually display Complete Shrinkage or Latent Decoupling. (ii) When using Latent Scaling to identify finetuning-only latents, BatchTopK crosscoders consistently identify more such latents.
>
> These results demonstrate that our findings reproduce across narrow domain finetuning (medical & reasoning), different architectures (Qwen & Llama), and alternative finetuning algorithms (RL tuning), supporting the generality and robustness of our analysis. We will add these results to the Appendix and make sure to properly refer to them in the main text.

---

> > ### Comment · Reviewer_A8m4 · 2025-08-04
> >
> > Thank you for the detailed response, I hope the next version will benefit from all the edits! I maintain my (positive) rating of 5. It is not lower because the paper correctly identifies an issue with current methods and proposes a reasonable fix. It is not higher because even though this fix works, it still feels ad hoc and targeted towards addressing the issues of one specific method (even after adding the analysis for other fine-tuning settings). Excited for the follow up study!

---

### Decision · Program_Chairs · 2025-09-17

**Decision:**

Accept (poster)

**Comment:**

This paper advances model diffing by showing that widely used L1‑trained crosscoders systematically over‑declare chat‑only concepts due to "Complete Shrinkage" and "Latent Decoupling". This is demonstrated by the proposed Latent Scaling test and KL‑based causal interventions. The recommendation to use BatchTopK crosscoders is well‑motivated: under BatchTopK, Delta‑norm becomes a reliable proxy for causal importance, whereas it fails under L1. While novelty may be moderate (adapting BatchTopK and scaling ideas to crosscoders), the contribution corrects a methodological pitfall and provides a simple, effective remedy.